*Report*

**EMBO** *reports*

# FBXO38 is dispensable for PD-1 regulation

Nikol Dibus[1], Eva Salyova [iD][2], Karolina Kolarova[1], Alikhan Abdirov [iD][1,3], Michele Pagano [iD][4✉], Ondrej Stepanek [iD][2✉] & Lukas Cermak [iD][1✉]

## Abstract

**SKP1-CUL1-F-box protein (SCF) ubiquitin ligases are versatile protein complexes that mediate the ubiquitination of protein substrates. The direct substrate recognition relies on a large family of F-box-domain-containing subunits. One of these substrate receptors is FBXO38, which is encoded by a gene found mutated in families with early-onset distal motor neuronopathy. SCF^FBXO38 ubiquitin ligase controls the stability of ZXDB, a nuclear factor associated with the centromeric chromatin protein CENP-B. Loss of FBXO38 in mice results in growth retardation and defects in spermatogenesis characterized by deregulation of the Sertoli cell transcription program and compromised centromere integrity. Moreover, it was reported that SCF^FBXO38 mediates the degradation of PD-1, a key immune-checkpoint inhibitor in T cells. Here, we have re-addressed the link between SCF^FBXO38 and PD-1 proteolysis. Our data do not support the notion that SCF^FBXO38 directly or indirectly controls the abundance and stability of PD-1 in T cells.**

**Keywords** PD-1; FBXO38; Cullin; Immune Checkpoint; Protein Degradation
**Subject Categories** Cancer; Immunology; Post-translational Modifications & Proteolysis

## Introduction

Cullin-RING ubiquitin ligases (CRLs) constitute a major group of enzymes responsible for targeting various proteins for ubiquitin- and proteasome-dependent degradation (Skaar et al, 2013). These multi-subunit E3 ubiquitin ligases have separate modules for substrate recognition and catalysis. Many of their substrate-binding receptors are dysregulated in various human diseases, including cancer and neurodegenerative syndromes (Skaar et al, 2014).

F-box-containing protein 38 (FBXO38), also known as MoKA, is a substrate receptor for one of many CRL1s (CUL1-RING ligases), also known as SCFs (for SKP1, CUL1, F-box protein ubiquitin ligases) (Jin et al, 2004; Smaldone et al, 2004). Several FBXO38 gene mutations were found in patients with early-onset distal hereditary motor neuronopathy (Sumner et al, 2013). Initially, FBXO38 was discovered as a modulator of the transcription factor Krüppel-like factor 7 (KLF7) activity (Smaldone et al, 2004), but its role as a CRL1 substrate receptor was poorly characterized. Then, the programmed cell death 1 (PD-1, also known as PDCD1, CD279) protein involved in immune-checkpoint control was identified as its substrate (Meng et al, 2018). Subsequently, SCF^FBXO38 was shown to play a role in centromeric chromatin control by targeting two zinc finger proteins, ZXDA and ZXDB, for degradation both in vivo and in vitro, impacting the stabilization of CENP-B protein at centromeres (Dibus et al, 2022a; Dibus et al, 2022b).

As these discoveries pointed to a potentially interesting crosstalk between different cellular processes occurring in different sub-cellular domains, we decided to investigate the role of FBXO38 in the control of PD-1 and connect it to its nuclear functions. Importantly, beside the original observation (Meng et al, 2018), there was no conclusive information in literature to support the role of FBXO38 in PD-1 biology. A comprehensive CRISPR/ Cas9-based genomic-wide screen exploring positive and negative regulators of PD-1 protein expression identified several known ubiquitin ligases potentially linked to PD-1 ubiquitination and degradation, including SCF substrate receptors FBXO28 and FBXO47, as well as the Cullin-independent membrane-localized ubiquitin ligase RNF43 (Okada et al, 2017). Notably, FBXO38 did not emerge as a hit in this screen, a result consistent with its absence in other screens focused on PD-1 protein or the T-cell exhaustion pathway (Belk et al, 2022; Dong et al, 2019). Moreover, the FBXO38-dependent degron composition identified in our previous study was confirmed through large-scale ubiquitin ligase degron identification (Dibus et al, 2022a; Zhang et al, 2023). Yet, the primary sequence in the intracellular domain of PD-1 did not contain any similar motifs. Additionally, individuals harboring FBXO38 mutations exhibit neurodegenerative syndromes without concomitant immune deficiencies or alterations in the immune system (Akcimen et al, 2019; Sumner et al, 2013).

While the non-identification of FBXO38 in above-mentioned screens does not definitively rule out its role as a PD-1 regulator, it aligns with the findings of our targeted approach. Our attempts to reproduce key experimental outcomes that substantiate SCF^FBXO38 function as a ubiquitin ligase for the PD-1 receptor were unsuccessful. Instead, our results indicate that SCF^FBXO38 is not involved in PD-1 regulation.

[1]Laboratory of Cancer Biology, Institute of Molecular Genetics of the Czech Academy of Sciences, Prague, Czech Republic. [2]Laboratory of Adaptive Immunity, Institute of Molecular Genetics of the Czech Academy of Sciences, Prague, Czech Republic. [3]Faculty of Science, Charles University, Prague, Czech Republic. [4]Department of Biochemistry and Molecular Pharmacology, Howard Hughes Medical Institute, Laura and Isaac Perlmutter NYU Cancer Center, New York University Grossman School of Medicine, New York, NY 10016, USA. ✉E-mail: michele.pagano@nyulangone.org; ondrej.stepanek@img.cas.cz; lukas.cermak@img.cas.cz

# Results and discussion

## FBXO38 and the PD-1 receptor localize to distinct subcellular compartments

To validate the findings by previous study (Meng et al, 2018), we performed a partial replication involving PD-1 overexpression. However, upon PD-1 overexpression in HEK293FT cells (lacking endogenous expression of PD-1 protein), we observed a significant accumulation of PD-1 in the cytoplasm (Fig. 1A; white arrowheads).

To minimize such overexpression artifacts, we utilized a Sleeping Beauty transposon-based system to generate HEK293FT-PD-1. The addition of doxycycline induced human PD-1 expression at levels comparable to or lower than those in T cells. In this optimized system, approximately $10^6$ copies of *hPDCD1* mRNA are expressed per 1 µg of total RNA (Appendix Table S1). In contrast, highly expressed genes like *ACTB* or *GAPDH*, are consistently expressed at levels between $10^7$ and $10^8$ copies of mRNA per 1 µg (Glare et al, 2002).

Moreover, instead of using tagged PD-1, we optimized the use of a commercially available antibody to detect PD-1 protein in western blots and paraformaldehyde-fixed cells (Fig. 1B,C). This allowed us to determine whether FBXO38 and PD-1 are located in the same cellular compartments. As with endogenous PD-1, doxycycline induction led to PD-1 localization predominantly at cellular membranes, including the plasma membrane (Fig. 1C) (Pentcheva-Hoang et al, 2007). Our previous studies demonstrated that FBXO38 is localized in the nucleus both in cultured cells and in vivo (Dibus et al, 2022a; Dibus et al, 2022b). Exogenous FBXO38 localizes to the nucleus when not overexpressed above the physiological level, which is a consequence of its strong nuclear localization signal (Fig. 1D). To test whether PD-1 expression affects FBXO38 localization or whether PD-1 could be stabilized in the nucleus, we detected both FBXO38 and PD-1 by immuno-fluorescence upon inhibition of the proteasome (with MG-132) or inhibition of neddylation (using MLN4924, which prevents SCF-mediated degradation) (Duda et al, 2008). In contrast to plasma membrane-associated PD-1, endogenous FBXO38 was strictly nuclear. Incubation with MG-132 or MLN4924 did not lead to colocalization of FBXO38 and PD-1 in any subcellular compartment (Fig. 1E).

## FBXO38 neither associates with the PD-1 receptor nor influences its protein levels

Next, we tested the interaction between FBXO38 and PD-1. We pretreated the cells with MLN4924 to avoid the degradation of PD-1, which could be potentially caused by FBXO38 overexpression. Consistent with our previous observations (Dibus et al, 2022a; Dibus et al, 2022b), FBXO38 interacted with its substrate ZXDB, and deletion of either its C-terminus or its F-box motif disrupted this interaction (Fig. 2A). However, we did not detect any interaction between PD-1 and FBXO38. To verify that inducible PD-1 correctly folds and associates with its established interactors (Okazaki et al, 2001), we immunopurified it from HEK293FT-PD-1 cells. Immunoprecipitated PD-1 interacted with its canonical partner SHP2, but not with FBXO38 or SKP1 (Fig. 2B).

The discrepancy between our findings and those of Meng et al, in the analysis of the FBXO38:PD-1 interaction may stem from variations in experimental protocols. Overexpression of large nuclear proteins can lead to mislocalization, as observed in previous studies (Bolognesi and Lehner, 2018). Additionally, SCF substrate receptors may associate with potential substrates post-lysis (Reitsma et al, 2017), potentially yielding false positive results when proteins, which cannot interact in vivo due to being localized into different cellular compartments, meet after cell lysis. Even mild concentrations of non-ionic detergents, such as NP-40 or Triton X-100, can cause that nuclear proteins not tightly associated with chromatin intrude into cytosolic and membrane fractions. To circumvent these artifacts, reliable separation methods, such as mechanical cell disruption in hypotonic conditions or digitonin-based lysis, are essential. The use of turbonuclease in all our experiments effectively prevented nonspecific interactions potentially caused by nucleic acid contamination. Furthermore, given the absence of the FBXO38:PD-1 interaction, we found it unnecessary to replicate the in vitro ubiquitination reactions performed in the previous study.

Subsequently, we evaluated the impact of protein degradation inhibition on PD-1 protein levels. We observed that they remained unchanged following the inhibition of SCF or the proteasome with MLN4924 and MG-132, respectively (Fig. 2C). In contrast, the protein level of p27, an established substrate of $SCF^{FBXL1}$, increased (Carrano et al, 1999). The previous study showed that over-expression of FBXO38 in HEK293FT cells leads to increased PD-1 degradation (Meng et al, 2018). This contrasts with our previous observation that FBXO38 overexpression does not enhance the degradation of its substrates (Dibus et al, 2022a). Accordingly, we observed only a modest decrease in ZXDB protein levels and no changes in PD-1 levels upon FBXO38 overexpression. However, inhibition of the SCF-dependent degradation led to an increase in FBXO38, ZXDB, and p27 levels, with no effects on PD-1 protein levels (Fig. 2D).

We also monitored the levels of ZXDB and PD-1 in control and FBXO38-depleted cells with or without cycloheximide treatment to inhibit protein synthesis. We observed a substantial increase in the protein levels and stability of ZXDB, but not PD-1 in FBXO38 knocked-down cells (Fig. 2E). Expression of an FBXO38 construct restored ZXDB degradation, but exhibited no effect on PD-1 protein levels.

Quantifying PD-1 protein through western immunoblotting presents a challenge due to its significant posttranslational modifications. Studies indicate the importance of PD-1 fucosylation (N-type glycosylation) for proper trafficking and plasma membrane localization (Okada et al, 2017). Additionally, recent findings describe substantial O-glycosylation in the stalk region of PD-1 (Tit-Oon et al, 2023). As shown in Fig. 1B, these glycosylation types are not prevalent in HEK293FT cells. However, for experiments investigating PD-1 in T cells, potential post-lysis deglycosylation using PNGase is necessary. PNGase selectively removes specific N-type modifications, not O-type, possibly resulting in multiple PD-1 variants with distinct molecular weights. To ensure methodological precision, we adhered to established field standards, employing flow cytometry analysis with specific antibodies, as demonstrated in the following results. This approach alleviated potential selectivity issues associated with the incomplete deglycosylation of specific PD-1 protein fractions.

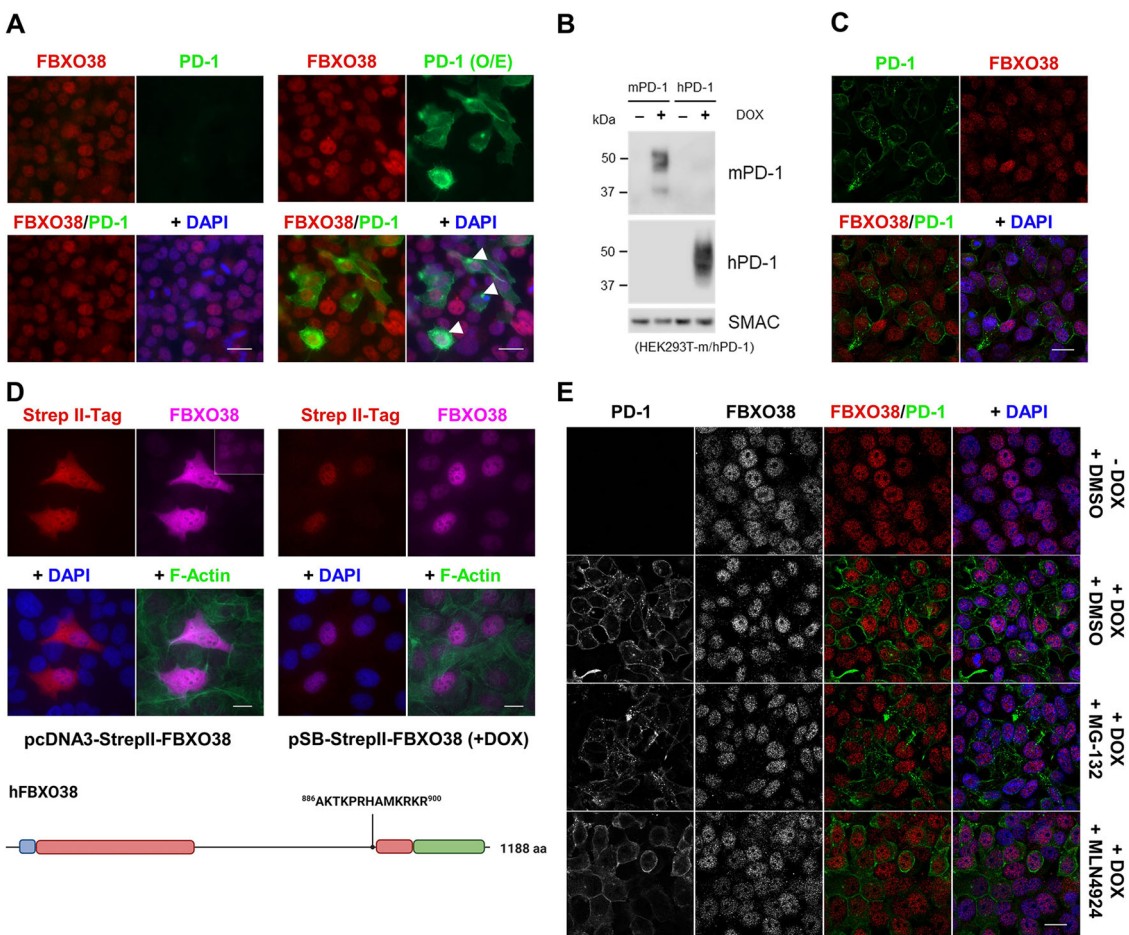

**Figure 1. FBXO38 and the PD-1 receptor localize to distinct subcellular compartments.**

(A) HEK293FT cells were transfected with a pcDNA3.1 vector expressing human PD-1. The cells were fixed and permeabilized 48 h after transfection and subsequently incubated with FBXO38 and PD-1 antibodies. DNA was stained with DAPI. White arrowheads mark PD-1 cytosolic accumulation. Scale bar, 20 μm. (B) HEK293FT-mPD-1 and HEK293FT-hPD-1 cells were incubated with doxycycline, lysed, and immunoblotted as indicated. SMAC staining was used as a loading control. (C) HEK293FT cells with the inducible expression of human PD-1 (HEK293FT-hPD-1) were incubated with doxycycline for 48 h and subsequently fixed, permeabilized, and incubated with anti-FBXO38 and PD-1 antibodies. DNA was stained with DAPI. Scale bar, 20 μm. (D) HEK293FT cells were transfected with pcDNA3.1 or pSB vectors expressing human FBXO38. The cells were fixed and permeabilized 48 h after transfection and subsequently incubated with FBXO38 and Strep II-tag antibodies. Prior to fixing, cells transfected with pSB vector were treated for 16 h with doxycycline. DNA was stained with DAPI. Polymerized actin was visualized using phalloidin. The intensity of overexpressed FBXO38 (pCDNA3), both Strep II tag and FBXO38 staining, was reduced to facilitate a comparison of localization. In the upper right corner, the staining from FBXO38 shows an image subset with comparable intensity. Scale bar, 20 μm. In the lower panel, the schematic representation of the structure of FBXO38 is depicted. Following the F-box motif (blue), the structured leucine repeat-rich region (red) is divided into two parts by the unstructured serine- and threonine-rich regions. The C-terminal part, which is involved in substrate binding, is colored green. The position of the nuclear localization signal (NLS) is shown above. (E) HEK293FT-hPD-1 cells were incubated with doxycycline and then treated with MG-132 or MLN4924. Cells were fixed 6 h after treatments, permeabilized, and incubated with FBXO38 and PD-1 antibodies. DNA was stained with DAPI. Scale bar, 20 μm. Source data are available online for this figure.

## FBXO38 does not alter PD-1 receptor membrane levels or stability in T cells

Next, we assessed the proposed role of SCF$^{FBXO38}$ in regulating PD-1 levels in T-cell leukemia cell lines and primary mouse T cells. First, we induced the expression of endogenous PD-1 in Jurkat and HPB-ALL cell lines using ionomycin and phorbol myristyl acetate (PMA) in the presence or absence of proteasome and neddylation inhibitors. Flow cytometry analysis showed that upon activation, the majority of the cells exhibited strong surface PD-1 expression (Fig. EV1A,B). Proteasome inhibition by MG-132 had no significant effect in Jurkat cells, but increased the percentage of

PD-1-positive cells and the level of PD-1 surface expression in the HPB-ALL cell line (Figs. 3A–D and EV1A,B). The inhibition of neddylation did not induce an increase in the PD-1 surface expression and the percentage of PD-1 positive cells in both cell lines (Figs. 3A–D and EV1A,B), which contradicts the hypothesis that PD-1 is degraded via SCF ubiquitin ligases and other CRLs, whose activity is neddylation-dependent. On the contrary, it resulted in decreased expression, suggesting a potential interplay between CRL-dependent degradation and other CRL-independent mechanisms controlling PD-1 protein levels. On the other hand, we do not dispute the regulation of PD-1 protein via proteasome-based degradation since we also found that the inhibition of the

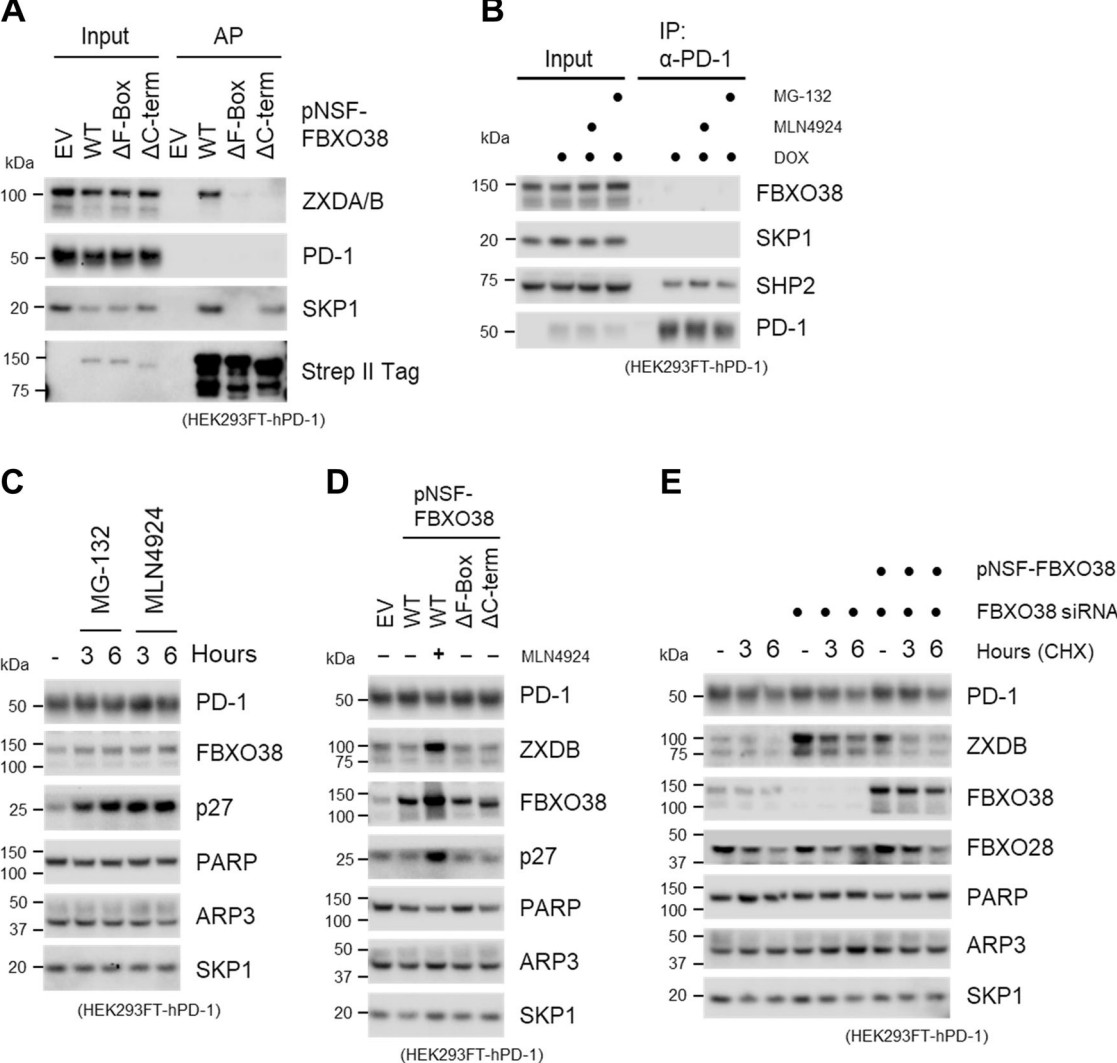

**Figure 2. FBXO38 neither associates with the PD-1 receptor nor regulates its protein levels.**

(A) HEK293FT-hPD-1 cells were transfected with Strep II-FLAG-tagged WT FBXO38 (pNSF-FBXO38 WT), FBXO38 lacking the F-box motif (ΔF-Box) or C-terminus (ΔC-term). The cells were treated with MLN4924 inhibitor 6 h prior to collecting, lysed, and subjected to affinity purification (AP) using Strep-Tactin resin and immunoblotted as indicated. SKP1 or ZXDB staining was used as the positive control. Inputs represent 1% of the whole-cell lysates subjected to AP. (B) HEK293FT-hPD-1 cells were treated with doxycycline, MG-132, and MLN4924 where stated. Whole-cell lysates were subjected to immunoprecipitation (IP) and immunoblotted as indicated. SHP2 staining was used as a positive control. Inputs represent 1% of the whole-cell lysate subjected to IP. (C) HEK293FT-hPD-1 cells were treated with MLN4924 and MG-132 for 3 or 6 h, lysed, and immunoblotted as indicated. p27 staining was used as a positive control, and PARP, ARP3, and SKP1 as the loading controls. (D) HEK293FT-hPD-1 cells were transfected with Strep II-FLAG-tagged WT FBXO38 (pNSF-FBXO38 WT), FBXO38 lacking the F-box motif (ΔF-Box) or C-terminus (ΔC-term). Where stated, the cells were treated with MLN4924 inhibitor 6 h prior to collection. Cells were lysed and immunoblotted as indicated. ZXDB and p27 stainings were used as positive controls, and PARP, ARP3 and SKP1 as loading controls. (E) HEK293FT-hPD-1 cells were transfected with non-targeting siRNA or a mixture of three different siRNAs targeting FBXO38 along with an empty pcDNA3.1 vector or SF-FBXO38 WT vector. Cells were then treated with cycloheximide (CHX) for the indicated time. Whole-cell lysates were immunoblotted as indicated. ZXDB and FBXO28 stainings were used as positive controls, and PARP, ARP3, and SKP1 as loading controls. Source data are available online for this figure.

proteasome leads to an increased level of membrane-bound PD-1 protein in the HPB-ALL T-cell line.

To assess the impact of FBXO38 deficiency on PD-1 expression in primary T cells, we employed our established *Fbxo38* knockout (KO) mouse model. While the *Fbxo38* mRNA is expressed in this mouse strain, a CRISPR-mediated deletion targeting the F-box motif led to the generation of the strain incapable of producing functional SCF[FBXO38] ubiquitin ligase (Dibus et al, 2022b). First, we assessed the immune system of the *Fbxo38*[KO/KO] mouse by

immunophenotyping of splenocytes. Our analysis encompassed a comprehensive assessment of T cells, B cells, NKT cells, monocytes, eosinophils, and neutrophils, yet, no significant differences were noted within these cell populations between the wild-type (WT) and *Fbxo38*[KO/KO] mice (Fig. EV1C). Intriguingly, NK-cell percentages consistently exhibited a modest decrease in *Fbxo38*[KO/KO] splenocytes. While this observation may hold interest within the context of FBXO38 role in the immune system, it is unrelated to PD-1 regulation in T cells.

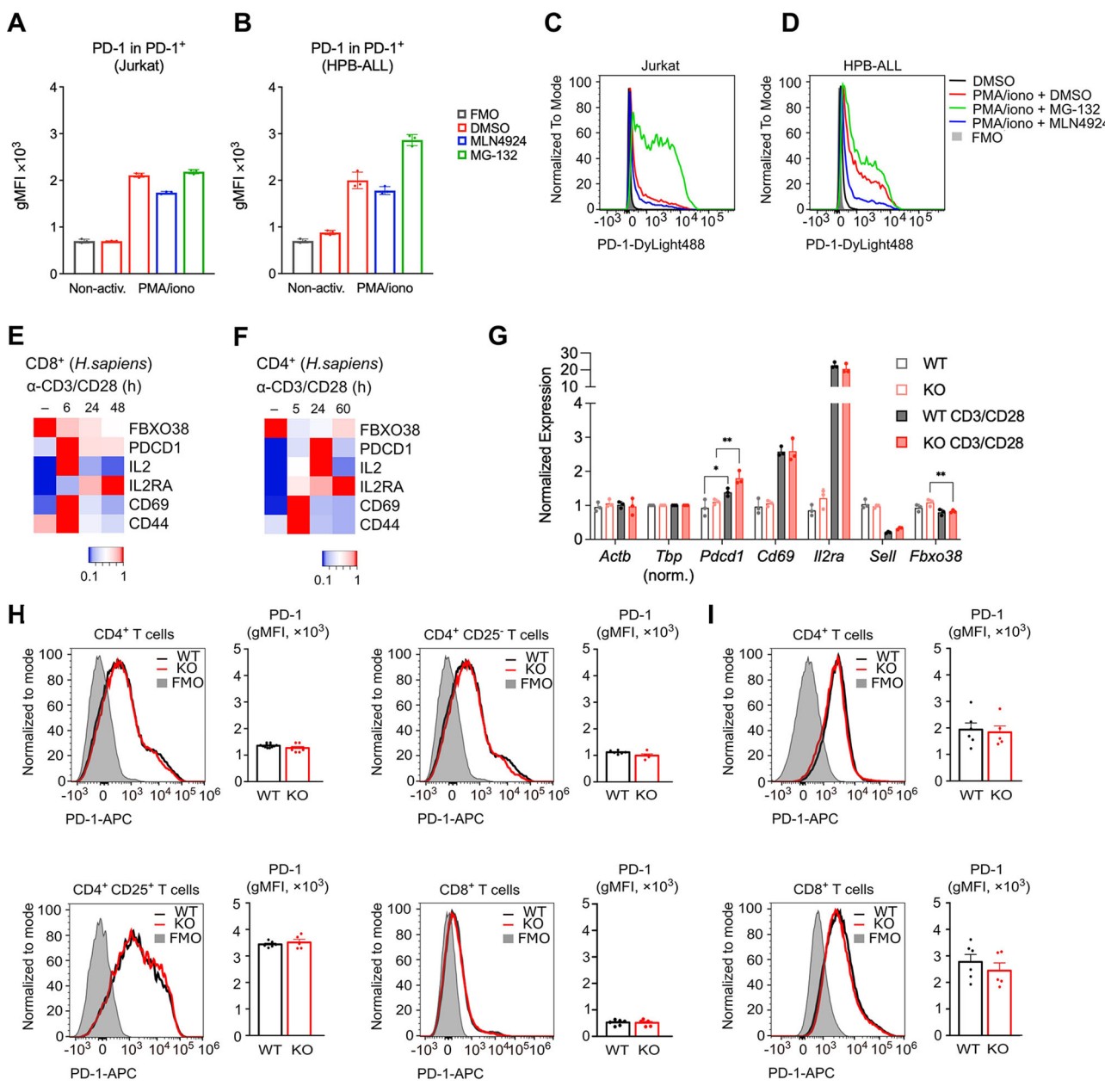

**Figure 3. FBXO38 does not control the levels and stability of PD-1 in T cells.**

(A–D) Jurkat (A) or HPB-ALL (B) T-cell leukemia cell lines were incubated for 72 h with phorbol myristyl acetate (PMA) and ionomycin (iono) followed by 6 h treatments with MG-132 or MLN4924 ($n = 3$; biological replicates). Surface PD-1 expression was analyzed by flow cytometry. (A, B) Bars represent the geometric mean of PD-1 fluorescence intensities in the PD-1$^+$ population (gMFI + SD). (C, D) Histograms represent the analysis of representative samples. (E, F) Publicly available data of gene expression changes upon activation of CD8$^+$ T cells (E; GSE122149; mean from triplicate) (Data ref: Ogando et al, 2019a; Ogando et al, 2019b) and CD4$^+$ T cells (F; GSE116697; mean from duplicate) (Data ref: Yukawa et al, 2020a; Yukawa et al, 2020b) using anti-CD3 and anti-CD28 antibodies. Data are visualized as a heatmap showing gene expression normalized to the highest obtained count (or average count from replicates). Publicly available data were normalized using a GREIN application (http://www.ilincs.org/apps/grein/) (Mahi et al, 2019). (G) Relative expression of markers in B-cell-depleted splenocytes. mRNAs isolated from naive or activated T cells from WT and Fbxo38 KO animals ($n = 3$; biological replicates) were reverse transcribed and subjected to RT-qPCR analysis. Expression levels were normalized to Tbp gene expression. Error bars indicate SD, and each data point represents an individual animal. (H, I) Cell-surface expressions of PD-1 in T cells isolated from the spleens of Fbxo38$^{WT/WT}$ (WT) and Fbxo38$^{KO/KO}$ (KO) mice. Histograms of representative mice and the quantification of the geometric mean fluorescence intensities (gMFI) for all mice (mean + SEM) are shown. (H) Steady-state surface levels of PD-1 in naive CD4$^+$, CD4$^+$ CD25$^-$, CD4$^+$ CD25$^+$, and CD8$^+$ T cells. $n = 6$ Fbxo38$^{WT/WT}$ and 5 Fbxo38$^{KO/KO}$ mice in two independent experiments. (I) Surface levels of PD-1 in CD4$^+$ (upper) and CD8$^+$ T (lower) cells stimulated with anti-CD3/CD28 beads (1:1 bead-to-cell ratio) in the presence of IL-2 for 96 h. $n = 6$ Fbxo38$^{WT/WT}$ and 5 Fbxo38$^{KO/KO}$ mice in two independent experiments. Source data are available online for this figure.

Next, we analyzed T cells from WT and *Fbxo38^{KO/KO}* mice at the steady state and upon activation. First, we analyzed publicly available data and found that *FBXO38* is consistently down-regulated at the transcriptional level during the early time points following T-cell activation via CD3 and CD28 receptor engagement (Fig. 3E,F) (Data ref: Ogando et al, 2019a; Ogando et al, 2019b; Data ref: Yukawa et al, 2020a; Yukawa et al, 2020b). In contrast, *Pdcd1* (PD-1) mRNA levels mimicked other known markers of T cell activation *Cd44*, *Cd69*, *Il2ra*, and *Il2* (Fig. 3E,F). In both CD4^+ and CD8^+ T cells, *Pdcd1* level peaked early and declined during the second and third days of activation. Therefore, day four was selected as the most promising time point to monitor the putative effect of FBXO38 expression on membrane PD-1 levels. Moreover, this time point was also used in the previous study (Meng et al, 2018).

In the next step, we compared the response of WT and *Fbxo38^{KO/KO}* T cells to the anti-CD3/CD28 activation in the presence of IL-2. The RT-qPCR analysis revealed that the pattern of *Pdcd1, Cd69, Sell*, and *Il2ra* expression at 96 h post-activation was consistent with T-cell activation and comparable in both strains. The level of *Fbxo38* mRNA did not change significantly upon activation (Fig. 3G). We observed that a small fraction of activated T cells was still progressing through the cell cycle at 96 h post-activation with no difference between the WT and *Fbxo38^{KO/KO}* T cells (Fig. EV1D). These experiments revealed that there are no apparent differences between the response of WT and *Fbxo38^{KO/KO}* T cells to the activation, at least on the transcriptional level.

Finally, we analyzed PD-1 and FBXO38 levels in the WT and *Fbxo38^{KO/KO}* mice. At the steady state, PD-1 was expressed almost exclusively in regulatory CD4^+ CD25^+ T cells at similar levels in both strains (Figs. 3H and EV2A). Upon activation, the FBXO38 protein was upregulated (Fig. EV2B), suggesting that it is regulated post-translationally. The canonical SCF^{FBXO38} substrate ZXDB was upregulated in the activated *Fbxo38^{KO/KO}* T cells, which validated our experimental approach (Fig. EV2B). We hypothesize that ZXDB is ubiquitinated by another ubiquitin ligase in naive cells, which becomes inactivated upon T-cell receptor engagement. This results in increased degradation of FBXO38 through self-ubiquitination (de Bie and Ciechanover, 2011). Upon activation and entry into the cell cycle, FBXO38 likely becomes the primary ubiquitin ligase for ZXDB.

Ex vivo activation with anti-CD3/CD28 beads induced the upregulation of PD-1 surface levels in CD4^+ and CD8^+ T cells (Fig. 3I). However, *Fbxo38^{KO/KO}* T cells did not exhibit higher PD-1 surface levels than their WT counterparts, as it would have been expected if PD-1 was a substrate for SCF^{FBXO38}. If anything, the PD-1 surface levels were rather slightly lower in *Fbxo38^{KO/KO}* than in WT T cells (Fig. 3I).

To assess the potentially differential impact of FBXO38 deficiency on extracellular and intracellular PD-1, we employed flow cytometry to analyze cell-surface and total PD-1 levels in T cells at the steady state and upon activation. Analysis of differences between surface and whole-cell PD-1 staining revealed that there is a minor fraction of PD-1 in the intracellular compartment. However, we did not observe any increase in total PD-1 levels in *Fbxo38^{KO/KO}* T cells compared to WT (Fig. EV2C–E).

## FBXO38 does not alter PD-1 receptor membrane levels upon viral infection

Next, we analyzed how FBXO38 expression affects PD-1 levels in T cells responding to the lymphocytic choriomeningitis virus (LCMV) infection in vivo. First, we analyzed publicly available data from the time course of LCMV infection and found that *Pdcd1* mRNA begins to decline between the third and seventh days after infection (Fig. 4A) (Data ref: Zhai et al, 2021; Zhai et al, 2021). This decline is followed by a similar decrease in mRNA for *Ifng*, indicating the successful clearance of the virus and termination of the immune response. Furthermore, *Fbxo38* mRNA is slightly inhibited during the first three days, coinciding with the activation of *Il2ra* (Fig. 4A). For this reason, we decided to monitor the PD-1 levels on T cells on day eight post-infection. A relatively large fraction of CD4^+ and CD8^+ T cells upregulated PD-1 during the LCMV infection (Figs. 4B and EV3A–C). In agreement with the steady-state and ex vivo activation data, PD-1 surface levels on PD-1^+ CD4^+ or CD8^+ T cells were equal or slightly lower in *Fbxo38^{KO/KO}* than in WT mice (Fig. 4C). Using D^b-NP396 tetramer, we gated on LCMV-specific CD8^+ T cells. We observed a comparable frequency of D^b-NP396 in WT and *Fbxo38^{KO/KO}* mice (Fig. EV3C). These D^b-NP396 tetramer^+ CD8^+ T cells were almost all PD-1 positive. Similar to the previous results, we observed no differences in the surface PD-1 expression in CD8^+NP396 tetramer^+ T cells from WT and *Fbxo38^{KO/KO}* mice (Figs. 4C and EV3A,B).

In summary, our investigation revealed no upregulation of surface PD-1 levels on T cells from *Fbxo38* KO mice, both in steady-state conditions and following ex vivo activation or viral infection. Analyses of CD8^+ T cells, conventional CD4^+ T cells, and regulatory CD25^+ CD4^+ T cells did not support the notion that FBXO38 acts as a negative regulator of PD-1. Notably, Meng et al, employed conditional *Fbxo38* KO mice driven by *Cd4*-CRE, potentially contributing to the discrepancy between our results. However, we find this unlikely. First, detailed immunophenotyping of the *Fbxo38* KO mice showed that these mice do not exhibit a severe phenotype in the leukocyte populations. Accordingly, the frequency of virus-specific CD8^+ T cells was comparable in LCMV-infected WT and *Fbxo38* KO mice, indicating that the whole-body FBXO38 deficiency does not interfere with normal anti-viral immune response. Second, there is a hypothetical possibility of compensatory mechanisms in the T-cell compartment in the whole-body KO mice. However, it is unclear why these putative mechanisms would not apply as well in the *Cd4*-CRE-driven conditional KO mice, which deletes *Fbxo38* early in the T-cell development in the thymus.

A low amount of intracellular PD-1 within the T cells suggest that it undergoes either rapid degradation or recycling back to the plasma membrane. Crucially, our experiments demonstrated that the intracellular fraction of PD-1 remained unaffected by FBXO38 expression. While we cannot entirely rule out the possibility that a small subset of intracellular PD-1 protein is regulated by FBXO38 (e.g., in the nuclear compartment), our data strongly indicate that such regulation does not exert a significant impact on the overall abundance of PD-1 or its localization at the plasma membrane.

Meng et al, concluded that FBXO38-deficient T cells exhibit impaired response to IL-2 immunotherapy, which is based on the

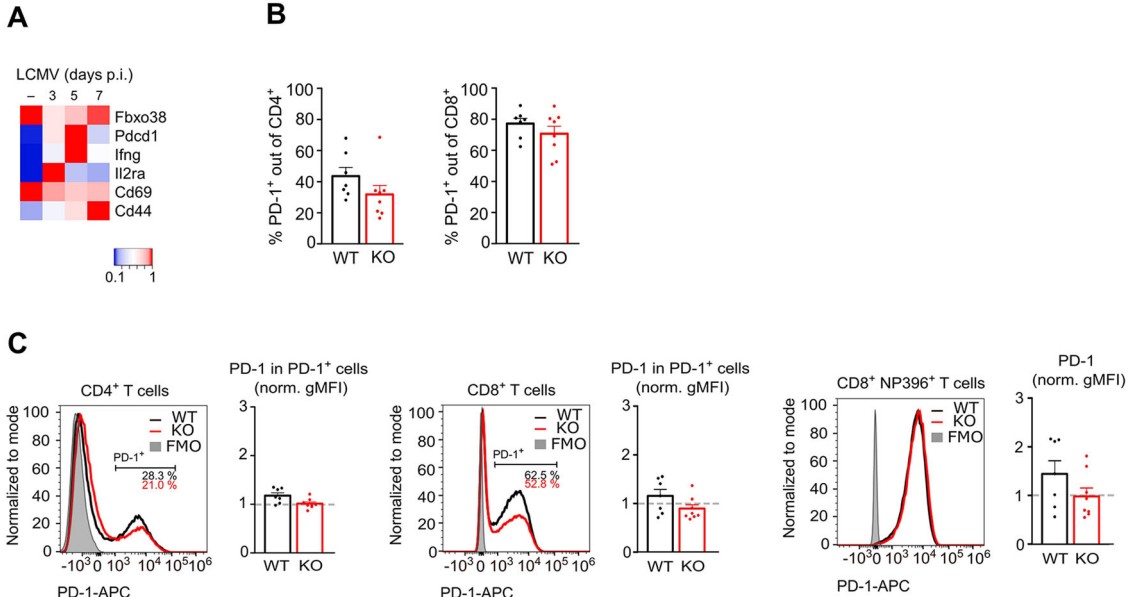

**Figure 4. FBXO38 does not regulate the levels of PD-1 upon viral infection.**

(A) Publicly available data of gene expression changes in CD8$^+$ T cells (GSE176310) in mice infected with LCM virus. Each time point represents the mean value of two different CD8$^+$ T cell isolations (GEO accession number GSE176310) (Data ref: Zhai et al, 2021; Zhai et al, 2021). Data are visualized as a heatmap showing gene expression normalized to the highest obtained mean from replicates. Publicly available data were normalized using the GREIN application (http://www.ilincs.org/apps/grein/) (Mahi et al, 2019). (B) Percentage of PD-1$^+$ cells out of CD4$^+$ and CD8$^+$ T cells from *Fbxo38*$^{WT/WT}$ and *Fbxo38*$^{KO/KO}$ mice infected with LCMV. The bars represent mean + SEM. $n = 7$ *Fbxo38*$^{WT/WT}$, 8 *Fbxo38*$^{KO/KO}$ mice in two independent experiments. (C) Cell-surface expressions of PD-1 in T cells isolated from the spleens of LCMV-infected *Fbxo38*$^{WT/WT}$ and *Fbxo38*$^{KO/KO}$ mice. The results of two independent experiments were normalized to the PD-1 gMFI of *Fbxo38*$^{WT/KO}$ T cells (set as 1). Histograms represent results from littermate mice. Bars show normalized cell-surface PD-1 levels in CD4$^+$ PD-1$^+$, CD8$^+$ PD-1$^+$ and CD8$^+$ D$^b$-NP396 tetramer$^+$ T cells (all PD-1$^+$ gate). $n = 7$ *Fbxo38*$^{WT/WT}$, 8 *Fbxo38*$^{KO/KO}$, and 8 *Fbxo38*$^{WT/KO}$ mice in two independent experiments. Individual experiments are shown in Fig. EV3. Source data are available online for this figure.

evidence that IL-2 inhibited tumor growth in 2 out of 10 WT mice, but not in *Fbxo38* cKO mice. However, the primary distinction between WT and *Fbxo38* cKO mice in their experiment appears to be the faster tumor growth in *Fbxo38* cKO mice. We do not challenge the general observations by Meng et al, that FBXO38-deficient T cells elicit impaired anti-tumor protection, which was shown in multiple experiments throughout their paper. However, the underlying mechanism is still to be elucidated as our data suggest that it does not involve the regulation of PD-1 levels by FBXO38. Notably, recent reports showed that FBXO38 positively controls the cGAS–STING pathway, crucial in regulating anti-tumor responses (Wu et al, 2024).

Despite the central role of the immune-checkpoint inhibitory receptor PD-1 in cancer immunotherapy, the regulation of the PD-1 levels and posttranslational modifications is incompletely understood. Meng et al, proposed an important mechanism of PD-1 regulation via SCF$^{FBXO38}$-mediated degradation. Our results challenge their observations on two different levels. First, we could not detect the interaction and colocalization of FBXO38 and PD-1 even when we pharmacologically inhibited the proteasome or neddylation, which is necessary for the SCF activity. Second, we could not observe any effect of FBXO38 on the levels and stability of PD-1 in vitro and in vivo.

Collectively, our data indicate that FBXO38 does not regulate the ubiquitination and overall levels of PD-1 receptor in T cells.

## Methods

### Methods and protocols

#### Cell culture procedures

Human cell line HEK293FT (ATCC #CRL-1573) was maintained in Dulbecco's Modified Eagle's Medium (DMEM) supplemented with 10% fetal bovine serum (FBS), and penicillin, streptomycin, and gentamicin, and cultured in a humidified incubator at 37 °C with 5% CO$_2$. Where indicated, cycloheximide (100 μg/ml), MLN4924 (1 μM; 6 h), MG-132 (10 μM; 6 h), or doxycycline hyclate (0.1 μg/ml; 48 h) were used. Cells were transfected with pSBtet containing PD-1 and the transposase-containing pSB100X using Lipofectamine 2000 (Invitrogen) according to the manufacturer's protocol. Positive clones were selected by puromycin 48 h after transfection. Transient transfections were carried out using polyethylenimine (PEI MW 25000, Polysciences). Jurkat E6 (ATCC #TIB-152) and HPB-ALL (DSMZ #ACC-483) were cultured in RPMI 1640 medium supplemented with 10% FBS and antibiotics. All cell lines were routinely cultured in the presence of potent anti-mycoplasma antibiotics (Plasmocure; Invivogen) and subsequently tested under laboratory conditions for the presence of mycoplasma using qPCR, along with the assessment of membranous DAPI positivity.

**Reagents and tools table**

| Reagent/resource | Reference or source | Identifier or catalog number |
| --- | --- | --- |
| 2-Mercaptoethanol | Sigma | M6250 |
| 4',6-diamidin-2-fenylindol (DAPI) | Sigma | D9542 |
| Benzonase | Santa Cruz | Sc-391121 |
| Cycloheximide | Sigma | C7698 |
| Desthiobiotin | IBA Lifesciences | 2-1000-025 |
| Dithiothreitol | Sigma | 10197777001 |
| Doxycycline hyclate | Sigma | D9891 |
| Formaldehyde | Thermo Fisher | 28908 |
| Gentamycin | Sandoz | |
| mIL-2 Recombinant Protein | Gibco | PMC0024 |
| Ionomycin calcium salt | Sigma | 10634 |
| Lipofectamine 2000 | Thermo Fisher | 11668019 |
| LIVE/DEAD fixable near-IR-dye | Thermo Fisher | L34976 |
| Methanol | Penta | 67-56-1 |
| MG132 | Medchemexpresss | HY-13259 |
| MLN4924 | Medchemexpresss | HY-70062 |
| Paraformaldehyde | Electron Microscopy Sciences | 15710 |
| Penicilin | BB Pharma | |
| Phorbol myristyl acetate (PMA) | Sigma | P1585 |
| Plasmocure | Invivogen | ant-pc |
| Polyethyleneimine (MW 25 K) | Polysciences | 23966 |
| Ponceau S | VWR | K793 |
| Propidium iodide | Sigma | 537059 |
| Protease Inhibitors Mini Tablets | Pierce | A32955 |
| Proteinase K | Sigma | P2308 |
| Puromycin | Sigma | P8833 |
| RNAiMax | Thermo Fisher | 13778150 |
| Sodium dodecyl sulfate | Sigma | 71736 |
| Sodium fluoride | Sigma | S7920 |
| Sodium orthovanadate | Sigma | 450243 |
| Streptomycin | Sigma | S9137 |
| Triton X-100 | Sigma | T8787 |
| **Experimental models** | | |
| *E. coli* | New England Biolabs | C2987I |
| HEK293FT | ATCC | CRL-1573 |
| HPB-ALL | DSMZ | ACC-483 |
| Jurkat E6 | ATCC | TIB-152 |
| C57BL/6N | Czech Centre for Phenogenomics | |
| LCMV (Armstrong) | European Virus Archive Global | |
| **Recombinant DNA** | | |
| **Expression vectors** | **Backbone** | **N-term. tag** |
| FBXO38 | pcDNA3.1 (pNSF) | 1xFLAG-2xStrepII |
| hPD-1 | pcDNA3.1 | |
| hPD-1 | pSB | |
| mPD-1 | pSB | |

| Reagent/resource | Reference or source | Identifier or catalog number | |
|---|---|---|---|
| **Sleeping beauty system** | **Vendor** | **Cat. number** | |
| FBXO38-containing plasmids | Dibus et al, 2022a | N/A | |
| pSBtet-Pur | Addgene | 60507 | |
| pSB100X | Addgene | 34879 | |
| **Antibodies** | | | |
| **Antigen** | **Vendor** | **Catalog number** | **RRID** |
| ARP3 | Santa Cruz | ab49671 | AB_2257830 |
| SH-PTP2 | Santa Cruz | sc-280 | AB_632401 |
| FBXO28 | Bethyl | A302-377A | AB_1907260 |
| FBXO38 | Atlas Antibodies | HPA041444 | AB_2677484 |
| FLAG | Cell Signaling | 14793 | AB_2572291 |
| p27 | Santa Cruz | sc-528 | AB_632129 |
| PARP-1 | Santa Cruz | sc-8007 | AB_628105 |
| SKP1 | Cell Signaling | 12248 | AB_2754993 |
| Strep II Tag | Novus Biologicals | NBP2-43735 | AB_2916323 |
| ZXDA/B | Atlas Antibodies | HPA043789 | AB_2678673 |
| mPD-1 | BioLegend | 109111 | AB_10613470 |
| hPD-1 | ExBio | 11-176 | AB_2687629 |
| Cyclin A | Pagano Lab | | |
| B220 | BioLegend | 103204 | AB_312989 |
| CD19 | BioLegend | 152402 | AB_2629714 |
| CD8α-BV421 | BioLegend | 100753 | AB_2562558 |
| CD4-BV650 | BioLegend | 100546 | AB_2562098 |
| TCRβ-FITC | BioLegend | 109206 | AB_313429 |
| CD25-PE | BioLegend | 102008 | AB_312857 |
| CD25-PE-Cy7 | BioLegend | 102016 | AB_312865 |
| PD-1-APC | BioLegend | 135209 | AB_2251944 |
| B220-BV510 | BioLegend | 103248 | AB_2650679 |
| Streptavidin, R-Phycoerythrin | Thermo Fisher | S866 | |
| Viability (SytoxBlue) | Thermo Fisher | S34857 | |
| Fc Block | BD Biosciences | 553142 | AB_394657 |
| CD4-BUV496 | BD Biosciences | 741050 | AB_2870665 |
| CD5-BV421 | BD Biosciences | 562739 | AB_2737758 |
| CD44-BV510 | Biolegend | 103043 | AB_2561391 |
| CD8a-BV605 | Biolegend | 100744 | AB_2562609 |
| GITR-BV711 | BD Biosciences | 563390 | AB_2738176 |
| Ly6G-BV785 | Biolegend | 127645 | AB_2566317 |
| CD45-FITC | Biolegend | 103108 | AB_312973 |
| CD11b-PerCPCy5.5 | Biolegend | 101228 | AB_893232 |
| CD161-PE | SONY | 1143540 | |
| CD19-PE-CF594 | BD Biosciences | 562291 | AB_11154223 |
| CD25 -PC7 | BD Biosciences | 552880 | AB_394509 |
| PD1-APC | Biolegend | 109111 | AB_10613470 |
| Ly6C-AF700 | BD Biosciences | 561237 | AB_10612017 |
| CD62L-APC-Cy7 | BD Biosciences | 560514 | AB_10611861 |

| Reagent/resource | Reference or source | Identifier or catalog number | |
|---|---|---|---|
| Anti-rabbit IgG (Alexa Fluor® 555) | Abcam | ab150070 | AB_2783636 |
| Anti-mouse anti-IgG (DyLight 488) | Thermo Fisher | 35503 | AB_1965946 |
| Anti-mouse IgG, HRP-linked | Cell signaling | 7076 | AB_330924 |
| Anti-rabbit IgG, HRP-linked | Cell signaling | 7074 | AB_2099233 |
| **Beads/resin** | **Vendor** | **Catalog number** | |
| Dynabeads® Mouse T-Activator CD3/CD28 | Thermo Fisher | 11453D | |
| Dynabeads Untouched Mouse CD8 Cells Kit | Thermo Fisher | 11417D | |
| Protein G (magnetic) | Dynabeads | 10004D | |
| Strep-Tactin® Superflow resin | IBA | 2-1206-025 | |
| **Oligonucleotides and other sequence-based reagents** | | | |
| **Cloning primers** | **Sequence** | **RE site** | |
| hPD-1 (FWD) | AAAGGCCTCTGAGGCCACCATGCAGATCCCACAGGCGCCCTGG | SfiI | |
| hPD-1 (RVS) | CTTGGCCTGACAGGCCTCAGAGGGGGCCAAGAGCAGTGTCC | SfiI | |
| mPD-1 (FWD) | AAAGGCCTCTGAGGCCACCATGTGGGTCCGGCAGGTACCCTGG | SfiI | |
| mPD-1 (RVS) | CTTGGCCTGACAGGCCTCAAAGAGGCCAAGAACAATGTCC | SfiI | |
| **CRISPR PCR verification primers** | **Sequence** | **Position** | |
| mFBXO38 (FWD) | GGCTGTTCTCCTCTTTGTG | Exon 3 | |
| mFBXO38 (REV) | TCCTTCCTCCTGCTAGACTCT | Exon 3 | |
| **RT-qPCR primers** | **Sequence** | | |
| hPD-1 (FWD) | CAGATCCCACAGGCGCCCTGG | | |
| hPD-1 (REV) | GAGGGGCCAAGAGCAGTGTCC | | |
| hRPL23A (FWD) | TACGATGCTTTGGATGTTGC | | |
| hRPL23A (REV) | GGCAGCTGGACTCAGTTTAGA | | |
| mActb (FWD) | CTAAGGCCAACCGTGAAAAG | | |
| mActb (REV) | ACCAGAGGCATACAGGGACA | | |
| mTbp (FWD) | GGCGGTTTGGCTAGGTTT | | |
| mTbp (REV) | GGGTTATCTTCACACACCATGA | | |
| mCd69 (FWD) | ACTGGAACATTGGATTGGGCT | | |
| mCd69 (REV) | CCCGTCAAGTTGAACCAGC | | |
| mPdcd1 (FWD) | TGCAGTTGAGCTGGCAAT | | |
| mPdcd1 (REV) | GGCTGGGTAGAAGGTGAGG | | |
| mIl2ra (FWD) | AGAACACCACCGATTTCTGG | | |
| mIl2ra (REV) | GGCAGGAAGTCTCACTCTCG | | |
| mSell (FWD) | GGTCATCTCCAGAGCCAATC | | |
| mSell (REV) | TCCATGGTACCCAACTCAGG | | |
| mFbxo38 (ALL ISO- FWD) | GGGGCATATTTCAGCGAGTA | | |
| mFbxo38 (ALL ISO- REV) | GGCTCTCCATTGACATCACA | | |
| mFbxo38 (ALL ISO- FWD) | CCAGGGAAACGGCAGTAAGT | | |
| mFbxo38 (ALL ISO- REV) | GTGGGAGTTGTTGCACCTCT | | |
| **Small interfering RNAs** | **Sequence** | **Catalog number** | |
| FBXO38#1 | GGGUGUAUUUCAGCGAGUAUU | | |
| FBXO38#2 | GGACUCGAUUGGUUGAUAUUU | | |
| FBXO38#3 | GAGCGAAGCUGUUUGAGUAUU | | |
| CTRL_#1 | AUGAACGUGAAUUGCUCAAUU | D-001210-04 | |
| CTRL_#2 | AUGUAUUGGCCUGUAUUAGUU | D-001210-03 | |

| Reagent/resource | Reference or source | Identifier or catalog number |
|---|---|---|
| **Chemicals, enzymes and other reagents** | **Vendor** | **Catalog number** |
| 4X Bolt™ LDS Sample Buffer | Invitrogen | B0007 |
| Albumin (BSA) Fraction V | PanReac AppliChem | 9048-46-8 |
| Amersham™ Hybond® P Western blotting membranes, PVDF | Merck | GE10600023 |
| Biotinylated MHC I monomer H-2Db-NP396 (FQPQNGQFI) | NIH Tetramer Core Facility | |
| Blotto, non-fat dry milk | Santa Cruz Biotechnology | sc-2324 |
| Buffer E | IBA | 2-1000-025 |
| DpnI | Thermo Fisher | FD1703 |
| DMEM high glucose | Sigma- Aldrich | D6429 |
| Fetal bovine serum (FBS) | Thermo Fisher | 10270106 |
| LightCycler 480 SYBR Green I master mix | Roche | 04887352001 |
| Maxima™ H Minus cDNA Synthesis Master Mix | Thermo Fisher | M1662 |
| NuPAGE™ 4-12% Bis-Tris gel | Invitrogen | NP0323BOX |
| PfuX7 DNA polymerase | Addgene | 182364 |
| RPMI 1640 medium | Gibco™ | 11875093 |
| SalI | Thermo Fisher | FD0644 |
| SfiI | Thermo Fisher | FD1824 |
| SuperSignal™ West Femto Maximum Sensitivity Substrate | Thermo Fisher | 34095 |
| WesternBright™ ECL HRP Substrate Kits | Advansta | 490005-016 |
| ProLong Gold Antifade Mountant | Invitrogen | P36930 |
| Pierce™ BCA Protein Assay Kits | Thermo Fisher | 23225 |
| RNeasy Plus Kit | Qiagen | 74134 |
| **Software** | **Vendor/link** | |
| FlowJo 10.6.2 | BD Biosciences | |
| GraphPad Prism 5.0 | GraphPad Software | |
| GREIN application | http://www.ilincs.org/apps/grein/ | |
| Heatmapper | www.heatmapper.ca | |
| ImageJ | https://imagej.net | |
| Image Lab | Bio-Rad | |
| Zen 2.3 | Zeiss | |
| LAS | Leica | |
| **Other (hardware)** | **Vendor** | |
| Axio Imager Zeiss 2 | EC Plan-Nefluar objectives | |
| LSRII | BD Biosciences | |
| *Cytek Aurora* | Cytek | |
| *FACSSymphony* | BD Biosciences | |
| TCS SP8 | Leica | |

### Plasmid construction

cDNAs of human and mouse *PDCD1* were amplified by PCR from human peripheral blood and mouse lymph node cDNAs, respectively. Prepared cDNAs were cloned into the pSBtet-Pur plasmid using SfiI restriction sites.

FBXO38-containing plasmids were created as described previously (Dibus et al, 2022a). Briefly, human FBXO38 cDNA (Origene #RC204380) was cloned into pcDNA3.1 containing N-terminal twinStrepII-FLAG-tag (NSF). For the F-box-lacking FBXO38 variant, PCR mutagenesis was carried out using PfuX7 DNA polymerase with subsequent DpnI digestion.

### Gene silencing

Small interfering RNAs (Sigma Aldrich) were transfected into the subconfluent HEK293FT cells using Lipofectamine RNAiMAX (Invitrogen) according to the manufacturer's protocol. Non-targeting siRNAs were used as negative controls.

### Immunoblotting

For whole-cell lysates, cells were washed with ice-cold PBS and lysed in a lysis buffer (150 mM NaCl, 50 mM Tris pH 7.5, 0.4% Triton X-100, 2 mM $CaCl_2$, 2 mM $MgCl_2$, 1 mM EDTA) in the presence of phosphatase a protease inhibitor cocktail (Pierce) and Benzonase nuclease (0.125 U/µl; Santa Cruz) for 30 min on ice. Lysates were mixed with an equal volume of 2% SDS in 50 mM Tris-HCl, pH 8, heated for 5 min at 95 °C, and cleared by centrifugation.

Protein concentration was determined by the BCA method (Thermo Fisher Scientific). Samples were prepared by mixing with Bolt™ LDS Sample Buffer (Thermo Fisher Scientific) supplemented with 10% β-mercaptoethanol (Sigma Aldrich), heated for 5 min at 95 °C, and then separated using NuPAGE™ 4–12% gradient Bis-Tris gel (Invitrogen). Afterward, the samples were transferred to PVDF membrane (Amersham), blocked with 5% non-fat milk (Chem-Cruz), and incubated with indicated antibodies diluted in 3% BSA (PanReac Applichem) in tris-buffered saline with Tween® 20 (TBST) overnight at 4 °C. The HRP-conjugated secondary antibodies (Cell Signaling) were diluted in 5% milk in TBS-T. The membranes were developed using WesternBright ECL (Advansta) or SuperSignal™ West Femto Maximum Sensitivity Substrate (Thermo Fisher Scientific).

### Protein affinity purification and immunoprecipitation

The cells were collected and lysed in the lysis buffer (150 mM NaCl, 50 mM Tris pH 7.5, 0.4% Triton X-100, 2 mM $CaCl_2$, 2 mM $MgCl_2$, 1 mM EDTA, supplemented with phosphatase and protease inhibitors) for 10 min on ice. For Strep-tagged FBXO38, the lysates cleared by centrifugation were incubated with Strep-Tactin® Sepharose resin (IBA Lifesciences). Purified proteins were then eluted by desthiobiotin using Buffer E (IBA Lifesciences) and subsequently prepared for immunoblotting as described above. For immunoprecipitation of PD-1 protein from $1 \times 10^7$ HEK293FT-PD-1 cells, lysates were incubated with the PD-1 antibody (1 µg; Exbio #11–176) and mixed with Dynabeads® Protein G. Parental HEK293FT cells were used as a negative control. Elution of immunoprecipitated proteins was carried out with 1× Bolt™ LDS Sample Buffer (Thermo Fisher Scientific), and samples were prepared for immunoblotting as described above.

### Immunocytochemistry

Cells were washed with PBS, fixed with 3% PFA in PBS for 20 min, permeabilized with 0.2% Triton X-100 in PBS for 10 min, and blocked for 1 h (3% BSA, 0.1% Triton X-100 in PBS). Incubation with indicated primary antibodies was carried out for 2 h at RT, followed by incubation with Alexa Fluor-conjugated secondary antibodies (Abcam) for 30 min at RT. DAPI was used to stain DNA. Slides were mounted with ProLong Gold Antifade Mountant (Invitrogen). Images were acquired either using Axio Imager Zeiss 2 (EC Plan-Nefluar objectives) and analyzed with ZEN 2.3 or ImageJ software. Confocal images were acquired using Leica TCS

SP8 Confocal Microscope (HC PL APO CS2 40×/1.30 OIL objective) and analyzed with LAS software.

### Mouse model

$Fbxo38^{KO/KO}$ mice were generated in a C57BL/6N background using the CRISPR/Cas9 genome-editing system as described (Dibus et al, 2022b). Shortly, the Cas9 protein and gene-specific sgRNAs (Integrated DNA Technologies) were used for zygote electroporation. The genome editing was confirmed in the founder mouse using PCR.

### Immunophenotyping

Single-cell suspensions were prepared from spleens using a 100µm cell strainer. RBCs were lysed with ACK buffer. In total, $2 \times 10^6$ cells were stained with 50 µl FACS buffer (0.5 mM EDTA, 1 M HEPES and 2% FBS in HBSS without $Mg^{2+}$ and $Ca^{2+}$) supplemented with 20% BD Horizon Brilliant Stain Buffer Plus, Fcblock (1:200) and fluorochrome conjugated antibodies. After 30 min, cells were washed with FACS buffer. SYTOX Blue (1:40,000) was added and 300,000 live leukocytes were acquired on Cytek Aurora spectral flow cytometer.

### In vitro activation assay

T cells were isolated from mouse splenocytes using erythrocyte lysis (ACK buffer), followed by $B220^+$ or $CD19^+$ B-cells depletion with specific antibodies and anti-rat magnetic beads (Invitrogen 11417D). In all, $2 \times 10^6$ of the remaining cells were stimulated with $2 \times 10^6$ anti-CD3/CD28 beads (Gibco, 11453D) in 5 ml of IMDM medium supplemented with 10% FBS, antibiotics, and 10 ng/ml IL-2 (Gibco) for 96 h at 37 °C and 5% $CO_2$.

### LCMV infection

Mice were infected intraperitoneally with LCMV (Armstrong) dose of $2 \times 10^5$ PFU. The spleens were harvested on day 8 post-infection. Splenocytes were depleted of erythrocytes by incubation in ACK buffer. The samples were analyzed using Cytek Aurora (Cytek) or FACSSymphony (BD Biosciences) flow cytometers.

### Flow cytometry analysis

For the analysis of mouse T cells, the following antibodies were used: anti-CD8α-BV421 (clone 53–6.7, BioLegend 100753, diluted 200×), anti-CD4-BV650 (clone RM4-5, BioLegend 100546, diluted 200×), anti-TCRβ-FITC (clone H57-597, BioLegend 109206, diluted 400×), anti-CD25-PE-Cy7 (clone PC61, BioLegend 102016, diluted 200×) or anti-CD25-PE (clone PC61, BioLegend 102008, diluted 200×), anti-PD-1-APC (clone 29F.1a12, BioLegend 135209, diluted 200×), anti-B220-BV510 (clone RA3-6B2, BioLegend 103248, diluted 200×). LIVE/DEAD fixable near-IR dye (Thermo Fisher Scientific, L34976) was used for the viability staining.

LCMV-specific $CD8^+$ T cells were identified by specific H-2d$^b$-NP396 tetramer staining (diluted 200×). The tetramers were produced by incubating biotinylated H-2D$^b$-NP396-PE (FQPQNGQFI) monomers (from NIH Tetramer Core Facility) with streptavidin-PE (Thermo Fisher Scientific, S866) at a molar ratio of 3:1. Streptavidin-PE was added in two doses with 1 h incubation on ice after each step. Staining was performed in 2% FBS, 2 mM EDTA, 0.1% sodium azide in PBS on ice for 30 min.

Fluorescence minus one (FMO) stained samples without anti-PD-1 antibody were used to gate PD-1$^+$ cells.

For the analysis of human T cell-leukemia cell lines, anti-PD-1 (clone EH12-2H7; Exbio 11-176, diluted 500×) in combination with anti-mouse IgG secondary (Thermo Fisher Scientific 35503; DyLight 488) was used. Propidium iodide (1 μg/ml in PBS; Sigma) was used for the viability staining.

To perform intracellular staining, activated and naive cells were first fixed with 3% paraformaldehyde (PFA) and then stained with T-cell markers (anti-CD4/8). Subsequently, cells were either directly stained with anti-PD-1 antibody or permeabilized with 0.5% Triton X-100 (TX-100). After permeabilization, cells were further stained with anti-PD-1 antibody or 4',6-diamidino-2-phenylindole (DAPI) for subsequent cell cycle analysis.

### cDNA preparation and RT-qPCR analysis

Synthesis of cDNA was performed using B-cell-depleted splenocytes that were either naive or activated as described in detail above. Total RNA (1 μg) isolated from these cells using the RNeasy Plus kit (Qiagen, 74134) was subjected to reverse transcription using Maxima™ H Minus cDNA Synthesis Master Mix (Thermo Fisher, M1662).

For RT-qPCR analysis, the LightCycler 480 SYBR Green I master mix (Roche; 04887352001) was used and gene expression was analyzed using the LC480 Real-Time PCR system. Biological triplicate analyses were performed for each condition and relative expression levels of target genes were determined using the $2^{(Ct^{Act} - Ct^{Naive})}$ method with means of normalized Ct values. TATA-binding protein (Tbp) cDNA levels were used for normalization. Detailed information regarding qPCR primers is given in the Reagents and Tools table.

### Ethics statement

All animal experiments were approved by the Resort Professional Commission for Approval of Projects of Experiments on Animals of the Czech Academy of Sciences, Czech Republic (protocols 115/2016 and AVCR 1667/2022 SOV II) and were in accordance with the Czech Act No. 246/1992 Coll. and European directive 2010/63/EU.

Animals were maintained in a controlled, specific pathogen-free environment at the Animal Facilities of the Institute of Molecular Genetics of the Czech Academy of Sciences, Czech Republic. Mice were provided food and water ad libitum and kept in the animal facility with a 12-h light-dark cycle.

### Quantification and statistical analysis

Graphs were generated using Prism version 10 (GraphPad Software). The quantity/densitometry of protein in western blots was analyzed using Image Lab software (Bio-Rad). Flow cytometry data were analyzed using FlowJo version 10.6.2 (BD Biosciences). A representative experiment out of three is shown for all western blots and immunofluorescence pictures. The geometric mean was used for the bar charts, and the error bars show the standard deviation or standard error mean. If not stated otherwise, all reported experiments were replicated at least twice in the laboratory. In mouse experiments, the sample size was estimated as a minimal number of mice which can provide conclusive results based on our previous experience. Experiments were not randomized because they were structured based on specific genotypes. Blinding was not

implemented, as the experiments involved objectively measured parameters using devices (such as FACS) that were independent of the researchers' subjective judgment. Student's t-test was employed to compare the means of the two groups in the RT-qPCR analysis, following confirmation that the assumptions of normality and equal variances were met.

Publicly available data from the GEO database were normalized using the GREIN application (http://www.ilincs.org/apps/grein/) (Mahi et al, 2019). A heatmap was created with Heatmapper (www.heatmapper.ca) (Babicki et al, 2016).

## Data availability

All original uncut blots images, Ponceau S stainings, western blots densitometry, normalized Ct and expression values, and immuno-fluorescence images files are available as a Source data associated with Figures and Expanded View Figures. All flow cytometry files and gating strategies were deposited in BioStudies under accession number S-BSST1326 and are accessible via the following link https://www.ebi.ac.uk/biostudies/studies/S-BSST1326. All other raw data are available upon request.

The source data of this paper are collected in the following database record: biostudies:S-SCDT-10_1038-S44319-024-00220-8.

## Peer review information

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

## Acknowledgements

LC was supported by Czech Health Research Council (AZV: NU21-08-00312) and Czech Science Foundation (GA24-10435S). OS was supported by National Institute of Virology and Bacteriology (Programme EXCELES, ID421 Project No. LX22NPO5103) - Funded by the European Union - Next Generation E, and Czech Science Foundation (GA22-18046S). AA was partially supported by the Charles University Grant Agency (393722). AA and KK were partially supported by Charles University Grant SVV 260637. We thank the Light Microscopy and Flow Cytometry Core Facility, Institute of Molecular Genetics of the Czech Academy of Sciences, for their support with the imaging presented herein. The authors used services of the Czech Centre for Phenogenomics at the Institute of Molecular Genetics supported by the Czech Academy of Sciences RVO 68378050 and by the project LM2023036 Czech Centre for Phenogenomics and CZ.02.1.01/0.0/0.0/18_046/0015861 provided by Ministry of Education, Youth and Sports of the Czech Republic. Particularly, we thank Jana Balounova for help with immunophenotyping strategy. In addition, we thank Dr. Tomas Brdicka for providing leukemia cell lines and antibody against SHP2. Illustrations were created with BioRender.com.

## Author contributions

**Nikol Dibus**: Conceptualization; Investigation; Methodology; Writing—original draft; Writing—review and editing. **Eva Salyova**: Investigation. **Karolina Kolarova**: Investigation. **Alikhan Abdirov**: Investigation. **Michele Pagano**: Conceptualization; Supervision; Writing—review and editing. **Ondrej Stepanek**: Conceptualization; Supervision; Funding acquisition; Methodology; Writing—review and editing. **Lukas Cermak**: Conceptualization; Supervision; Funding acquisition; Investigation; Methodology; Writing—original draft; Writing—review and editing.

Source data underlying figure panels in this paper may have individual authorship assigned. Where available, figure panel/source data authorship is listed in the following database record: biostudies:S-SCDT-10_1038-S44319-024-00220-8.

## Disclosure and competing interests statement

MP is a scientific cofounder of SEED Therapeutics; receives research funding from and is a shareholder in Kymera Therapeutics; and is a consultant for, a member of the scientific advisory board of, and has financial interests in CullGen, SEED Therapeutics, Triana Biomedicines, and Umbra Therapeutics; however, no research funds were received from these entities, and the findings presented in this manuscript were not discussed with any person in these companies. The remaining authors declare no competing interests.

# Expanded View Figures

**Figure EV1. FBXO38 does not control the levels and stability of PD-1 in T cells.**

(A, B) Jurkat (left) or HPB-ALL (right) T-cell leukemia cell lines were incubated for 72 h with phorbol myristyl acetate (PMA) and ionomycin (iono) followed by 6 h treatments with MG-132 or MLN4924 ($n = 3$; biological replicates). Surface PD-1 expression was analyzed by flow cytometry. Graphs represent the percentage of PD-1-positive cells (A) and the geometric mean of PD-1 fluorescence intensities (gMFI + SD) from all cells (B). (C) Percentage of different lymphocytic populations out of CD45$^+$ cells from *Fbxo38$^{WT/WT}$* ($n = 4$; grey) and *Fbxo38$^{KO/KO}$* ($n = 3$; red) mice. The boxes represent the 25th and 75th percentiles. Statistical significance was assessed by an unpaired two-tailed *t*-test. (D) Cell cycle analysis of naive or activated CD4$^+$ (upper panels) and CD8$^+$ (lower panels) T cells from *Fbxo38$^{WT/WT}$* (WT) and *Fbxo38$^{KO/KO}$* (KO) mice. Cells were stained with DAPI and analysed by flow cytometry. The left and middle panels present three independent samples of each genotype with the FMO control. The right panel depicts the analysis of sub-G1 (apoptosis), G1, and G2-M phases in percentage. Source data are available online for this figure.

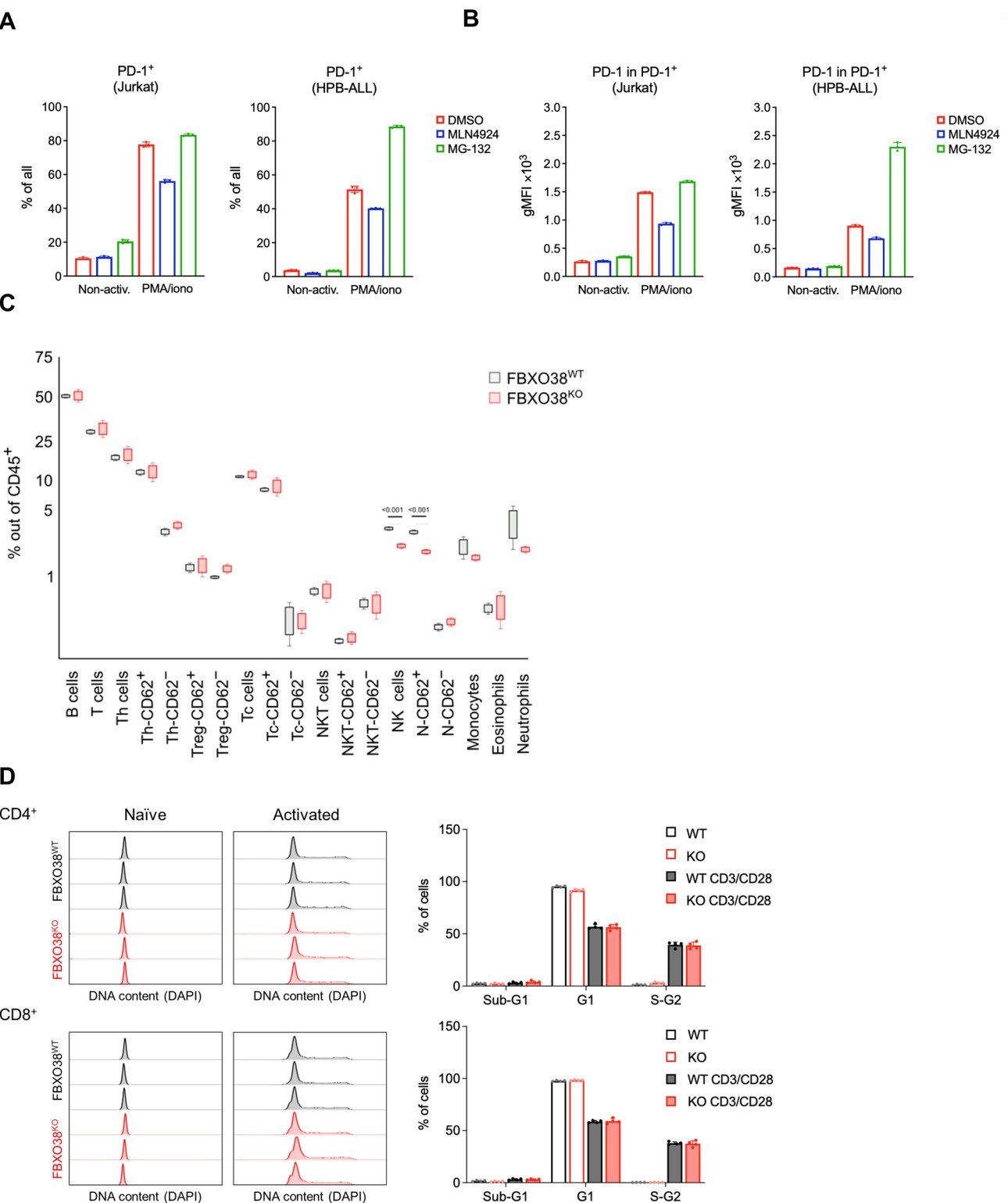

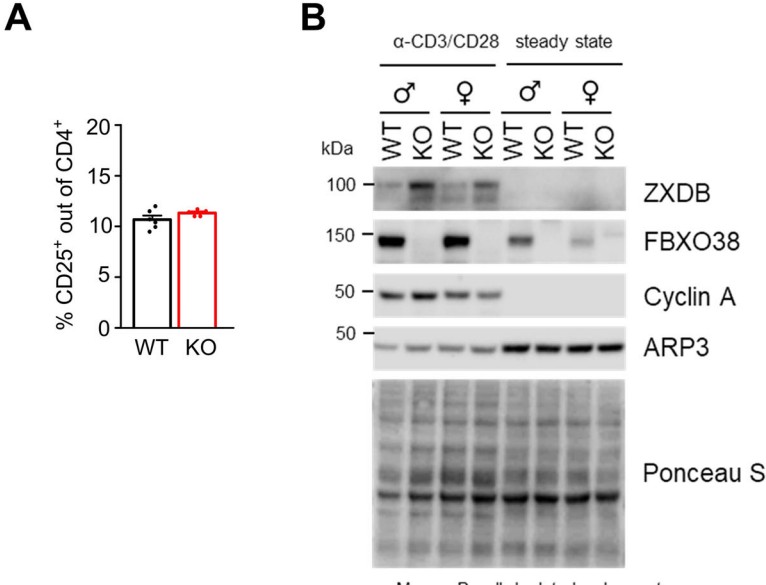

Figure EV2. **FBXO38 does not control the levels and stability of PD-1 in T cells.**

(A) Percentage of CD25$^+$ cells out of CD4$^+$ T cells. The bars represent mean + SEM. $n$ = 6 $Fbxo38^{WT/WT}$ (WT) and 5 $Fbxo38^{KO/KO}$ (KO) mice in two independent experiments (same as in Fig. 3H). (B) T cells isolated from mouse splenocytes (same as in Fig. 3H, I) were stimulated with anti-CD3/CD28 beads in presence of IL-2 for 96 h. Naive or anti-CD3/CD28 stimulated T cells were lysed and immunoblotted as indicated. ZXDB or cyclin A staining was used as the positive control and ARP3 as a loading control. Ponceau S staining was utilized to demonstrate protein loading. (C) Surface or whole-cell expression of PD-1 in naive or activated CD8$^+$ T cells from WT and Fbxo38 KO mice (same as in Fig. EV3D). Fixed cells were stained for surface markers (CD4 or CD8) and then either directly stained for PD-1 (surface) or alternatively permeabilized before PD-1 staining (whole-cell staining) as shown in scheme on the left panel. Samples were then analyzed by flow cytometry. Graphs represent the geometric mean of PD 1 fluorescence intensities (gMFI + SD) in CD8$^+$ T cells (left) or in PD-1$^+$ CD8$^+$ T cells (right). (D) Intracellular PD1 gMFI in naive or activated CD8$^+$ T cells from WT and Fbxo38 KO mice (same as in Fig. EV3G) as a result of difference between whole-cell and surface staining. (E) Surface or whole-cell expression of PD-1 in naive or activated CD4$^+$ (upper panel) and CD8$^+$ (lower panel) T cells from WT and Fbxo38 KO mice. Histograms show representative animal. The right panel shows three independent staining in comparison with naive cells.

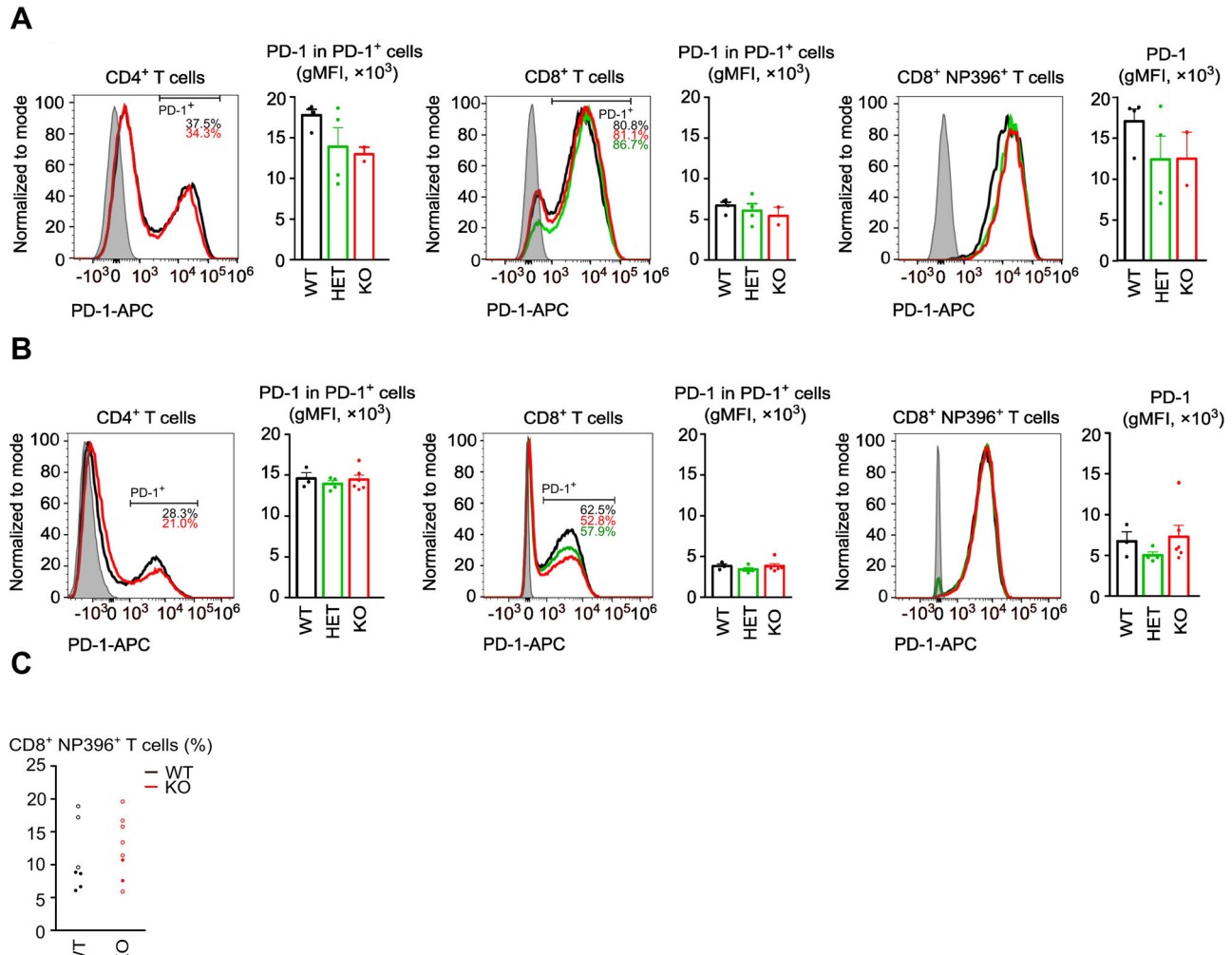

**Figure EV3. Fbxo38 does not control the levels or stability of PD-1 upon viral infection.**

(**A, B**) Cell-surface expressions of PD-1 in T cells isolated from the spleens of LCMV-infected *Fbxo38*^WT/WT and *Fbxo38*^KO/KO mice. The results of two independent experiments (**A, B**) are presented as normalized data in Fig. 4C. Histograms represent the analysis of representative samples. Bars show cell-surface PD-1 levels in CD4+ PD-1+, CD8+ PD-1+and CD8+ D^b-NP396 tetramer+ T cells (all PD-1+ gate). $n = 7$ *Fbxo38*^WT/WT, 8 *Fbxo38*^KO/KO, and 8 *Fbxo38*^WT/KO mice. (**C**) WT and KO mice were infected with LCMV. The percentage of Db-NP396 tetramer+ LCMV-specific T cells out of splenic CD8+ T cells was quantified 8 days post-infection by flow cytometry (same experiments as shown in Fig. 4C).

