## [Peer Review File · EMBO Reports]

FBXO38 is dispensable for PD-1 regulation

Nikol Dibus, Eva Salyova, Alikhan Abdirov, Karolina Kolarova, Michele Pagano, Ondrej Stepanek, and Lukas Cermak

Corresponding authors: Lukas Cermak (lukas.cermak@img.cas.cz) , Ondrej Stepanek (ondrej.stepanek@img.cas.cz), Michele Pagano (Michele.Pagano@nyulangone.org)

Review Timeline:

Submission Date:	7th Sep 23
Editorial Decision:	14th Nov 23
Revision Received:	31st Jan 24
Editorial Decision:	23rd Feb 24
Revision Received:	5th Mar 24
Accepted:	13th Jun 24

Transaction Report:

Dear Dr. Cermak

Thank you for the submission of your research manuscript to our journal. I apologize for the delay in handling your manuscript, but we have only now received the final referee report (see all three reports copied below).

As you will see, the referees acknowledge that your manuscript provides substantial evidence that PD-1 is not a primary target of FBXO38 in T cells. That said, the referees have also a number of suggestions how to further strengthen your data. Given that your manuscript presents negative data that are in disagreement with an earlier study, it is essential to address these concerns and to further substantiate your conclusions. Please also consider the suggestion from referee 2 to discuss alternative scenarios that, e.g., a minor fraction of PD-1 and particularly the internalized portion, might be targeted by FBXO38. Please ensure that you discuss your data in relation to the data from Meng et al in the most careful and appropriate manner, maybe also taking into account a potential impact of specific treatment protocols and cell context regarding the interaction and turnover of PD-1.

Given these constructive comments, we would like to invite you to revise your manuscript with the understanding that the referee concerns (as detailed above and in their reports) must be fully addressed and their suggestions taken on board. Please address all referee concerns in a complete point-by-point response. Acceptance of the manuscript will depend on a positive outcome of a second round of review. It is EMBO Reports policy to allow a single round of revision only and acceptance or rejection of the manuscript will therefore depend on the completeness of your responses included in the next, final version of the manuscript.

We realize that it is difficult to revise to a specific deadline. In the interest of protecting the conceptual advance provided by the work, we recommend a revision within 3 months (February 15th). Please discuss the revision progress ahead of this time with the editor if you require more time to complete the revisions.

I am also happy to discuss the revision further via e-mail or a video call, if you wish.

***** IMPORTANT NOTE:

We perform an initial quality control of all revised manuscripts before re-review. Your manuscript will FAIL this control and the handling will be delayed IN CASE the following APPLIES:

- 1) A data availability section providing access to data deposited in public databases is missing. If you have not deposited any data, please add a sentence to the data availability section that explains that.
- 2) Your manuscript contains statistics and error bars based on $n=2$. Please use scatter blots in these cases. No statistics should be calculated if $n=2$.

When submitting your revised manuscript, please carefully review the instructions that follow below. Failure to include requested items will delay the evaluation of your revision.*****

- 1) a .docx formatted version of the manuscript text (including legends for main figures, EV figures and tables). Please make sure that the changes are highlighted to be clearly visible.
- 2) individual production quality figure files as .eps, .tif, .jpg (one file per figure). Please download our Figure Preparation Guidelines (figure preparation pdf) from our Author Guidelines pages <https://www.embopress.org/page/journal/14693178/authorguide> for more info on how to prepare your figures.
- 3) a .docx formatted letter INCLUDING the reviewers' reports and your detailed point-by-point responses to their comments. As part of the EMBO Press transparent editorial process, the point-by-point response is part of the Review Process File (RPF), which will be published alongside your paper.
- 4) a complete author checklist, which you can download from our author guidelines (<<https://www.embopress.org/page/journal/14693178/authorguide>>). Please insert information in the checklist that is also

reflected in the manuscript. The completed author checklist will also be part of the RPF.

5) Please note that all corresponding authors are required to supply an ORCID ID for their name upon submission of a revised manuscript (<<https://orcid.org/>>). Please find instructions on how to link your ORCID ID to your account in our manuscript tracking system in our Author guidelines (<<https://www.embopress.org/page/journal/14693178/authorguide#authorshipguidelines>>)

6) We replaced Supplementary Information with Expanded View (EV) Figures and Tables that are collapsible/expandable online. A maximum of 5 EV Figures can be typeset. EV Figures should be cited as 'Figure EV1, Figure EV2' etc... in the text and their respective legends should be included in the main text after the legends of regular figures.

7) Please note that a Data Availability section at the end of Materials and Methods is now mandatory. In case you have no data that requires deposition in a public database, please state so instead of refereeing to the database. See also < <https://www.embopress.org/page/journal/14693178/authorguide#dataavailability>>. Please note that the Data Availability Section is restricted to new primary data that are part of this study.

8) At EMBO Press we ask authors to provide source data for the main figures. Our source data coordinator will contact you to discuss which figure panels we would need source data for and will also provide you with helpful tips on how to upload and organize the files. If you wish to deposit source data on external repositories, this is also possible.

Additional information on source data and instruction on how to label the files are available <<https://www.embopress.org/page/journal/14693178/authorguide#sourcedata>>.

10) Figure legends and data quantification:
The following points must be specified in each figure legend:

- the name of the statistical test used to generate error bars and P values,
 - the number (n) of independent experiments (please specify technical or biological replicates) underlying each data point,
 - the nature of the bars and error bars (s.d., s.e.m.)
- If the data are obtained from n {less than or equal to} 5, show the individual data points in addition to the SD or SEM.
- If the data are obtained from n {less than or equal to} 2, use scatter blots showing the individual data points.

See also the guidelines for figure legend preparation:
<https://www.embopress.org/page/journal/14693178/authorguide#figureformat>

11) Our journal encourages inclusion of *data citations in the reference list* to directly cite datasets that were re-used and obtained from public databases. Data citations in the article text are distinct from normal bibliographical citations and should directly link to the database records from which the data can be accessed. In the main text, data citations are formatted as follows: "Data ref: Smith et al, 2001" or "Data ref: NCBI Sequence Read Archive PRJNA342805, 2017". In the Reference list, data citations must be labeled with "[DATASET]". A data reference must provide the database name, accession number/identifiers and a resolvable link to the landing page from which the data can be accessed at the end of the reference. Further instructions are available at <<https://www.embopress.org/page/journal/14693178/authorguide#referencesformat>>.

12) All Materials and Methods need to be described in the main text. We would encourage you to use 'Structured Methods', our new Materials and Methods format. According to this format, the Materials and Methods section should include a Reagents and

Tools Table (listing key reagents, experimental models, software and relevant equipment and including their sources and relevant identifiers) followed by a Methods and Protocols section in which we encourage the authors to describe their methods using a step-by-step protocol format with bullet points, to facilitate the adoption of the methodologies across labs. More information on how to adhere to this format as well as downloadable templates (.doc or .xls) for the Reagents and Tools Table can be found in our author guidelines: <<https://www.embopress.org/page/journal/14693178/authorguide#manuscriptpreparation>>. An example of a Method paper with Structured Methods can be found here: <<https://www.embopress.org/doi/10.15252/msb.20178071>>.

13) As part of the EMBO publication's Transparent Editorial Process, EMBO Reports publishes online a Review Process File to accompany accepted manuscripts. This File will be published in conjunction with your paper and will include the referee reports, your point-by-point response and all pertinent correspondence relating to the manuscript.

Yours sincerely,

Referee #1:

The manuscript by Dibus et al entitled "SCF fbx038 is dispensable for PD-1 regulation" challenges a previously published paper that reported just the opposite results. Bearing in mind this, I have very carefull reviewed the data and experiments from this manuscript. Apart for minor mistakes (suchs as Fig 1e at the end of the first Results sections which should be 1f), the manuscript is:

1. Excellently written, presented and reasoned.
2. The authors take into account the pontential artifacts that could be derived from each technique.
3. The figures are of excellent quality, and the experiments clean and credible.
4. Their results are consistent with some experimental observations in our lab regarding PD-1 expression.
5. The authors use a range of experimental systems, from cell lines to the use of primary cells, wild type and from KO mice. With FBX038 expression, silenced expression and KO.

I believe that if there are discrepancies between published paper and data, the papers that cannot reproduce data should be published as well. This is critical for the scientific process and credibility of scientific data. Obviously, the authors from the original report (which is a Nature paper, I believe) should have a change to comment on it.

The experiments are well-conducted. However, due to their significance, I would propose an experiment (if it is technically possible) that would reinforce the conclusions from the authors. I would suggest to introduce a FBX038-specific target sequence into the cytoplasmic domain of PD-1, and co-express this molecule with FBX038 in some of the experimental conditions that the authors have used in the paper. This would show that if PD-1 would really contain a FBX038 target, PD-1 expression would be regulated by it. This experiment may not be required, but it would be something that I would personally do (if technically possible, of course)

Otherwise, I have no other comments but to recomend the publication of the paper.

Referee #2:

Summary:

Prior research has shown that conditional deletion of the SCF substrate receptor Fbxo38 resulted in accelerated tumor progression in mice due to elevated levels of PD-1 in T cells infiltrating the tumor. Treatment with anti-PD-1 therapy normalized

the effect of Fbxo38 deficiency on tumor growth in mice, indicating that PD-1 could be a target of FBXO38 in T cells (Meng et al).

In this manuscript, Dibus and colleagues have revisited the connection between SCF-FBXO38 and the degradation of PD-1 through a comprehensive approach that includes biochemical analysis, cell biology and genetic experiments, including the use of a whole-body Fbxo38-deficient (KO) mouse model. The authors present substantial evidence suggesting that PD-1 protein may not be a primary target of FBXO38 in T cells. This finding holds significance for our comprehension of PD-1 regulation and its potential connection to the FBXO38 pathway.

While the authors statement is evident, and their data is convincing, it might be wise not to entirely rule out the possibility that FBXO38 could target a small fraction of PD-1. I encourage the authors to consider moderating their conclusions and comments across the manuscript.

Specific comments:

- 1) The authors should remove unnecessary and incorrect statements in the results section. For example, in the manuscript they state that most of the biochemical experiments performed by Meng et al. are based on the overexpression of both FBXO38 and PD-1 in HEK293FT cells.
- 2) Please remove the conclusive statements from the figure legend text.
- 3) Fig 1. Indicate where PD-1 is overexpressed (Fig 1A) and supply images with higher resolution for better clarity, the current stains appear to have low resolution. The authors used a Sleeping beauty transposon-based system to induce PD-1 with doxycycline - to similar levels found in T-cells, can they provide comparable immunoblots of PD-1 expression from T-cells? They use proteasome and neddylation inhibitors and conclude that this does not alter the subcellular distribution of FBXO38 or PD-1 (or their co-localization) referring to Fig 1e. I assume it should be Fig 1f?
- 4) Fig 2. The biochemical findings are clear and straightforward, indicating that FBXO38 and PD-1 do not appear to bind when these proteins are expressed in HEK293 cells under basal culture conditions. They also demonstrate that knockdown of FBXO38 does not alter PD-1 protein levels, referencing Fig 1e. I assume it should be Fig 2e?
- 5) Meng et al. conducted a series of biochemical assays using PHA-stimulated T-cells to demonstrate the regulation of PD-1 ubiquitination through overexpression, knockdown, or knockout of FBXO38. They utilize a surface-labeling method to illustrate that internalized PD-1 undergoes ubiquitination and proteasome-dependent degradation, while PD-1 at the cell surface exhibit minimal ubiquitination in activated T cells. Presumably, internalization and ubiquitination are rapid and dynamic processes, possibly influenced by glycosylation and/or other PTMs which makes it is challenging to rule out the possibility that a fraction of FBXO38 does not interact with internalized PD-1. For instance, Meng et al deglycosylate PD-1 for better detection of the FBXO38-PD-1 interaction. This could be mentioned and discussed better.
- 6) Suppl Fig 3c. The authors state that the NK-cell percentages consistently exhibited a modest elevation in KO splenocytes, but according to the figure panel, NK cells appear to be reduced in the KO?
- 7) Suppl Fig 3e. Provide clarification in the legend regarding T-cell stimulation: Were T-cells stimulated for 72 hours using anti-CD3/CD28? The authors conclude that Fbxo38 is consistently inhibited at the transcriptional level after T-cell activation via CD3 and CD28 receptor engagement, but the Fbxo38 protein level is high compared to "steady state". According to Fig 3e, f, PD-1 is induced, can the authors provide additional immunoblot data of PD-1 expression in Fig S3e?
- 8) Fig 3h. The concentration of anti-CD3/CD28 used for ex vivo activation of T cells is not specified in the legend text. Have the authors tried to assess PD-1 surface levels in WT and KO cells after stimulation with different concentrations of anti-CD3/CD28 and at various time points?
- 9) In general, the flow cytometry plots and gating in Suppl Fig 3 are very challenging to read and axes should be clearly labeled with bigger font and text.
- 10) Fig 4. The authors' conclusion is based on the analysis of PD-1 protein levels eight days after LCMV infection in WT and KO cells. Please rephrase the subtitle as PD-1 protein stability was not examined. Again, could the authors provide additional data on PD1 surface levels in WT and KO cells at additional time points?
- 11) Meng et al demonstrated that both Fbxo38 mRNA and protein levels increase in T cells when subjected to CD3/CD28 stimulation along with IL-2. They conclude that IL-2 not only promote T cell activation and increase the proportion of PD-1+ cells, but also triggers the expression of Fbxo38, which in turn mediate degradation of PD-1 (in tumor-infiltrating CD8+PD-1+ T cells). As Dibus et al utilized LCMV infection as a model to investigate the impact of Fbxo38 on PD-1 levels it would be useful if the authors could provide data regarding the expression of Fbxo38 mRNA and protein in T cells stimulated by CD3/CD28 + IL-2.

Referee #3:

In the presented manuscript, Dibus and colleagues challenge the finding of another group regarding the involvement of the E3 ligase FBXO38 in the turnover and stability of the surface protein PD-1. Given the clinical significance of PD-1, our understanding of the biology and its control are important and need to be expanded to gather insights into its full potential and putative druggability.

The group has presented their negative data regarding the crosstalk between FBXO38 and PD-1 by repeating previously published (2018) experiments and extended these with their own mouse model and publicly available data.

The presented data is clear and compelling, however, additional points should be addressed to enhance our understanding and the implications of the new findings.

Abstract/Discussion:

Please share your thoughts regarding the significance of your finding, especially from a translational point of view.

Immunoblots throughout the manuscript:

Do the authors have access to additional loading control antibodies, such as ActB, Tub, GAPDH or Vinculin? Arp3 ought to be a membrane marker and is regulated during cell mobility and or cell fate changes. This might/could become important at point of immune cell challenge/activation.

FBXO38 ko mice: Could the authors please add IHC/IF of Fbxo38, PD-1 and ZXDB in immune cells of wild type and ko mice? This would further strengthen the reported observation.

FBXO38-PD-1 and cancer:

Meng et al. reported an involvement of FBXO38 in the T-cell mediated antitumour immunity. Could this be a discriminator for the biology of Fbxo38 towards PD-1 stability? Please clarify or discuss this point.

Overall, the presented manuscript is of high interest and touches on the regulation of a protein of high significance.

Response to reviewers' comments

We were thrilled that all reviewers were positive on the manuscript as a whole. We also appreciate the constructive and thoughtful suggestions made by the three reviewers, and we believe that the resulting additions and changes have enhanced the clarity and overall message of our study. Finally, we extend our gratitude to the three reviewers for their invaluable time and commitment to elevating the quality of this paper.

Reviewer #1 (*italicized in blue*):

The manuscript by Dibus et al entitled "SCF fbx038 is dispensable for PD-1 regulation" challenges a previously published paper that reported just the opposite results. Bearing in mind fthis, I have very carefull reviewed the data and experiments from this manuscript. Apart for minor mistakes (suchs as Fig 1e at the end of the first Results sections which should be 1f), the manuscript is:

- 1. Excellently written, presented and reasoned.*
- 2. The authors take into account the pontential artifacts that could be derived from each technique.*
- 3. The figures are of excellent quality, and the experiments clean and credible.*
- 4. Their results are consistent with some experimental observations in our lab regarding PD-1 expression.*
- 5. The authors use a range of experimental systems, from cell lines to the use of primary cells, wild type and from KO mice. With FBX038 expression, silenced expression and KO.*

I believe that if there are discrepancies between published paper and data, the papers that cannot reproduce data should be published as well. This is critical for the scientific process and credibility of scientific data. Obviously, the authors from the original report (which is a Nature paper, I believe) should have a change to comment on it.

The experiments are well-conducted. However, due to their significance, I would propose an experiment (if it is technically possible) that would reinforce the conclusions from the authors. I would suggest to introduce a FBX038-specific target sequence into the cytoplasmic domain of PD-1, and co-express this molecule with FBX038 in some of the experimental conditions that the authors have used in the paper. This would show that if PD-1 would really contain a FBX038 target, PD-1 expression would be regulated by it. This experiment may not be required, but it would be something that I would personallly do (if technically possible, of course)

Otherwise, I have no other comments but to recomend the publication of the paper.

We changed Fig. 1e to Fig. 1f.

Regarding the proposed experiment, we acknowledge the interesting nature of this idea. However, we would emphasize some considerations that might limit its feasibility and interpretation in the context of our study. It is established that degrons are short linear motifs that can be transferable to other proteins¹; thus, it may not be surprising if introducing an FBXO38-specific target sequence (i.e., the degron) into the cytoplasmic domain of PD-1 will make PD-1 a substrate of FBXO38. However, the FBXO38 degron identified by our laboratory is present within a zinc-finger structure. This was confirmed by large-scale FBXO38's degron mapping². Both labs showed that FBXO38 degron contains a tyrosine and a consecutive cysteine. Our unpublished data suggest that the tyrosine modification has impact on the FBXO38-dependent ubiquitination of cognate substrates. It would be unpredictable how such modifications are prevalent in the cellular membrane or endosomal environment. Importantly, it is crucial to note that PD-1 and FBXO38 localize to different cellular compartments. Thus, PD-1 levels may be unaffected despite the introduction of the FBXO38 degron due to the compartmental isolation between

PD-1 and FBXO38. In summary, the interpretation of both positive and negative results may produce confounding interpretations that might limit the clarity of the conclusions.

Once again, thank this Reviewer for his/her insightful and thoughtful suggestions.

Reviewer #2 (*italicized in blue*):

Summary:

Prior research has shown that conditional deletion of the SCF substrate receptor Fbxo38 resulted in accelerated tumor progression in mice due to elevated levels of PD-1 in T cells infiltrating the tumor. Treatment with anti-PD-1 therapy normalized the effect of Fbxo38 deficiency on tumor growth in mice, indicating that PD-1 could be a target of FBXO38 in T cells (Meng et al).

In this manuscript, Dibus and colleagues have revisited the connection between SCF-FBXO38 and the degradation of PD-1 through a comprehensive approach that includes biochemical analysis, cell biology and genetic experiments, including the use of a whole-body Fbxo38-deficient (KO) mouse model. The authors present substantial evidence suggesting that PD-1 protein may not be a primary target of FBXO38 in T cells. This finding holds significance for our comprehension of PD-1 regulation and its potential connection to the FBXO38 pathway.

While the authors statement is evident, and their data is convincing, it might be wise not to entirely rule out the possibility that FBXO38 could target a small fraction of PD-1. I encourage the authors to consider moderating their conclusions and comments across the manuscript.

Specific comments:

1) The authors should remove unnecessary and incorrect statements in the results section. For example, in the manuscript they state that most of the biochemical experiments performed by Meng et al. are based on the overexpression of both FBXO38 and PD-1 in HEK293FT cells.

We modified the text according to the reviewer's suggestion.

2) Please remove the conclusive statements from the figure legend text.

The conclusive statements were removed.

3) Fig 1. Indicate where PD-1 is overexpressed (Fig 1a) and supply images with higher resolution for better clarity, the current stains appear to have low resolution. The authors used a Sleeping beauty transposon-based system to induce PD-1 with doxycycline - to similar levels found in T-cells, can they provide comparable immunoblots of PD-1 expression from T-cells? They use proteasome and neddylation inhibitors and conclude that this does not alter the subcellular distribution of FBXO38 or PD-1 (or their co-localization) referring to Fig 1e. I assume it should be Fig 1f?

We have addressed the figure name errors. Concerning the resolution of Fig. 1a, we aimed to observe both cytosolic and surface PD-1, as well as nuclear FBXO38. Thus, we utilized a widefield fluorescence microscope. We suspect that the loss of resolution occurred during figure export. To rectify this, we have recreated the figure and included the original raw data for all microscope images. Additionally, we have clearly marked exogenously expressed PD-1 in Fig. 1a.

Regarding the expression levels of exogenous versus endogenous PD-1, we measured mRNA expression of doxycycline-induced PD-1 and compared it to the levels of PD-1 mRNA in activated T-cells. Detailed calculations and strategies are outlined in Appendix Table S1.

As for the western blot analysis of PD-1, we would like to emphasize that mature PD-1 protein in T-cells contains both N- and O-type glycosylation, and post-lysis deglycosylation is very challenging. In contrast, flow cytometry analysis of PD-1 expression, using antibodies that recognize non-glycosylated extracellular parts of PD-1 not directly involved in PD-1L binding, represents the best and most quantitative method to assess both extracellular and intracellular levels of PD-1.

4) Fig 2. The biochemical findings are clear and straightforward, indicating that FBXO38 and PD-1 do not appear to bind when these proteins are expressed in HEK293 cells under basal culture conditions.

They also demonstrate that knockdown of FBXO38 does not alter PD-1 protein levels, referencing Fig 1e. I assume it should be Fig 2e?

We have made the necessary corrections in the revised manuscript.

5) Meng et al. conducted a series of biochemical assays using PHA-stimulated T-cells to demonstrate the regulation of PD-1 ubiquitination through overexpression, knockdown, or knockout of FBXO38. They utilize a surface-labeling method to illustrate that internalized PD-1 undergoes ubiquitination and proteasome-dependent degradation, while PD-1 at the cell surface exhibit minimal ubiquitination in activated T cells. Presumably, internalization and ubiquitination are rapid and dynamic processes, possibly influenced by glycosylation and/or other PTMs which makes it is challenging to rule out the possibility that a fraction of FBXO38 does not interact with internalized PD-1. For instance, Meng et al deglycosylate PD-1 for better detection of the FBXO38-PD-1 interaction. This could be mentioned and discussed better.

To explore whether FBXO38 differentially regulates surface or cytosolic (internalized or ER/Golgi-associated) fractions of PD-1, we conducted additional experiments involving CD3/CD28 activation. T-cells were isolated from the spleens of four animals for each genotype and activated using CD3/CD28 beads (for 96 hours—consistent with Fig. 3g-h and the same time course used by Meng et al.). Quiescent or activated T-cells were fixed with paraformaldehyde, and PD-1 was detected in both permeabilized and non-permeabilized cells. We hypothesized that the differences between these stainings would unveil the intracellular portion of PD-1. The results of this experiment are presented in Figure EV1G-I, revealing no increase in PD-1 levels in FBXO38 knockout (KO) T-cells. Instead, in line with our previous findings, we observed rather a slight decrease in PD-1 levels in the KO animals. Correspondingly, we observed only a minimal increase in PD-1 protein staining in whole-cell flow cytometry analysis. This suggests that intracellular PD-1 protein is dynamically degraded or recycled back to the plasma membrane, supporting the significant role of proteasome-dependent degradation in PD-1 biology—a finding corroborated by our data (Fig. 3A-D). Notably, our data show no effect of FBXO38 expression on this dynamic process. Moreover, by assessing permeabilized cells, we examined the cell cycle progression of activated T-cells. Since PD-1 functions as an inhibitory molecule, actively blocking cell cycle progression when engaged with PD-L1 from adjacent T cells³, we observed that both wild-type (WT) and FBXO38 knockout (KO) T-cells progressed similarly. Although this experiment does not involve either recombinant PD-L1 or PD-L1 expressing APC cells, we believe that it further supports the idea that FBXO38 does not influence T-cell activation. The corresponding data have been incorporated into Figure EV1D.

All these new findings, including the potential regulation of other PD-1 fractions or the redundancy of FBXO38 degradation of PD-1 through other pathways, are discussed in the Discussion section.

7) Suppl Fig 3e. Provide clarification in the legend regarding T-cell stimulation: Were T-cells stimulated for 72 hours using anti-CD3/CD28? The authors conclude that Fbxo38 is consistently inhibited at the transcriptional level after T-cell activation via CD3 and CD28 receptor engagement, but the Fbxo38 protein level is high compared to “steady state”. According to Fig 3e, f, PD-1 is induced, can the authors provide additional immunoblot data of PD-1 expression in Fig S3e?

T cells were stimulated for 96 hours using anti-CD3/CD28 beads, and this information has been included in the legend. Additionally, we assessed the mRNA levels of *Fbxo38* after 96 hours of activation through qRT-PCR (Figure 3G). Our findings suggest that *Fbxo38* levels are only minimally, if at all, inhibited during this timeframe. *Fbxo38* mRNA levels appear to be inhibited after T-cell activation and subsequently recover. In contrast, increased protein levels of FBXO38 (after 96 hours of activation), as depicted in Figure EV1F, seem to result from the inhibition of FBXO38 protein degradation. We plan to investigate this observation in follow up studies.

As mentioned earlier, we chose to measure the protein level of PD-1 in T-cells using flow cytometry. This method was preferred over western blotting, given that T-cells exhibit extensive differential glycosylation and posttranslational modifications, making western blotting less advantageous. Crucially, Meng et al. utilized PNGase for deglycosylation to detect endogenous PD-1 in Jurkat cells, a method not employed in HEK-293FT—the cells used in our interaction studies. Recent findings strongly suggest significant O-glycosylation of PD-1 in the stalk region. Given that PNGase is specific only for certain N-type glycosylation, its application would likely result in hard-to-interpret outcomes⁴. Therefore, we prioritize the measurement of PD-1 levels using flow cytometry, which, as mentioned earlier, includes the assessment of the intracellular fraction.

8) Fig 3h. The concentration of anti-CD3/CD28 used for ex vivo activation of T cells is not specified in the legend text. Have the authors tried to assess PD-1 surface levels in WT and KO cells after stimulation with different concentrations of anti-CD3/CD28 and at various time points?

To ensure the recommended 1:1 cell-to-bead ratio for optimal T-cell stimulation, we employed 2×10^6 anti-CD3/CD28 beads for the activation of the 2×10^6 cells that remained after B-cell depletion. We included this information in the revised manuscript. This ratio is recommended by the manufacturer for T-cell stimulation. It is important to note that our focus was not on investigating PD-1 levels in response to varying concentrations of anti-CD3/CD28. Hence, we did not conduct experiments exploring PD-1 dynamics under different concentrations of stimulation. Moreover, we also attempted to reproduce the methods outlined by Meng et al. to mitigate discrepancies in reviewing the results. Crucially, by employing the standard concentration of CD3/CD28 beads along with IL-2, we observe both protein and gene expression markers of T-cell activation, along with functional outcomes such as cell cycle progression.

9) In general, the flow cytometry plots and gating in Suppl Fig 3 are very challenging to read and axes should be clearly labeled with bigger font and text.

We appreciate the reviewer's comments, and we sincerely apologize for any inconvenience caused by the difficulty in interpreting these figures. All flow cytometry plots depicted in the figures are in high-resolution, and the fonts used align with the recommendations outlined in the EMBO Reports guidelines. We believe that the comment is mostly focused on Supplementary Fig. 3k which represents the scheme of immunophenotyping. We have incorporated revised versions of these schemes (gating strategies) into the corresponding source data, and we hope that they will enhance clarity and facilitate easier comprehension.

10) Fig 4. The authors' conclusion is based on the analysis of PD-1 protein levels eight days after LCMV infection in WT and KO cells. Please rephrase the subtitle as PD-1 protein stability was not examined. Again, could the authors provide additional data on PD1 surface levels in WT and KO cells at additional time points?

We rephrased the subtitle in the revised version of the manuscript.

We decided to analyze the PD-1 levels in the LCMV infection on day 8 post infection, because at these later time points, the expression of PD-1 at the transcriptional level goes down (Fig. 4A), but there are still relatively high levels of PD-1 protein in LCMV-specific T cells (Fig. 4C). Moreover, the expression of FBXO38 is higher at the later time points than early time points after the infection (Fig. 4A). Thus, we believe that this time point is the ideal one to observe the eventual role of FBXO38 in the regulation of PD-1 stability. Since we did not observe any effect, we find it extremely unlikely that a regulation of PD1 by FBXO38 could be observed due to the high transcription of PD-1 and/or low expression of FBXO38 at early time points, and the low protein levels of PD-1 at late time points. We would like to emphasize that we complemented our day 8 LCMV analysis by the comparison of WT and FBXO38 KO T cells in the steady-state and after ex vivo activation (Fig. 3H,I). In all these experiments, there was no indication that FBXO38 regulates PD-1. For all these reasons, we believe that the financial cost and ethical aspects of

repeating these experiments with different time points is not justified. We have clarified this in the revised manuscript.

11) Meng et al demonstrated that both Fbxo38 mRNA and protein levels increase in T cells when subjected to CD3/CD28 stimulation along with IL-2. They conclude that IL-2 not only promote T cell activation and increase the proportion of PD-1+ cells, but also triggers the expression of Fbxo38, which in turn mediate degradation of PD-1 (in tumor-infiltrating CD8+PD-1+ T cells). As Dibus et al utilized LCMV infection as a model to investigate the impact of Fbxo38 on PD-1 levels it would be useful if the authors could provide data regarding the expression of Fbxo38 mRNA and protein in T cells stimulated by CD3/CD28 + IL-2.

We adopted the experimental setup utilized by Meng et al., incorporating IL-2 in all experiments. Following a 96-hour timeframe, we did not observe any increase in mRNA levels (Figure 3G). Notably, public data from various sources actually indicate a decrease in mRNA levels, particularly at earlier timepoints. Intriguingly, as mentioned and discussed above we noticed an increase in FBXO38 protein levels at the 96-hour timepoint. Follow up investigations will focus on understanding the factors influencing FBXO38 protein expression and elucidating the reasons for its protein stabilization at this stage of T-cell activation. It is noteworthy that this altered expression of FBXO38 does not impact PD-1 levels. However, it significantly affects levels of ZXDA protein, in agreement with its role as a substrate of FBXO38 *in vivo* and *in vitro*.

Once again, we are grateful for the insightful and valuable suggestions of this Reviewer.

Reviewer #3 (*italicized in blue*):

In the presented manuscript, Dibus and colleagues challenge the finding of another group regarding the involvement of the E3 ligase FBXO38 in the turnover and stability of the surface protein PD-1. Given the clinical significance of PD-1, our understanding of the biology and its control are important and need to be expanded to gather insights into its full potential and putative druggability.

The group has presented their negative data regarding the crosstalk between FBXO38 and PD-1 by repeating previously published (2018) experiments and extended these with their own mouse model and publicly available data.

The presented data is clear and compelling, however, additional points should be addressed to enhance our understanding and the implications of the new findings.

Abstract/Discussion:

Please share your thoughts regarding the significance of your finding, especially from a translational point of view.

While FBXO38 mutations have been implicated in neurodegenerative syndromes, we do not align with the perspective that it acts as a positive regulator of T-cell activation. Importantly, individuals from families with FBXO38 mutations do not exhibit signs of either immune response inhibition or overactivation⁵⁻⁷. Therefore, we do not believe that targeting FBXO38 represents a promising strategy that would potentiate the action of anti-tumor therapies based on immune checkpoint inhibition. On the other hand, our extensive study of this protein leads us to believe that it holds potential as a druggable target in male reproduction associated syndromes or distal neuromuscular atrophies^{6,8}. We mentioned this in the Discussion of the revised manuscript.

Immunoblots throughout the manuscript:

Do the authors have access to additional loading control antibodies, such as ActB, Tub, GAPDH or Vinculin? Arp3 ought to be a membrane marker and is regulated during cell mobility and or cell fate changes. This might/could become important at point of immune cell challenge/activation.

In response to this thoughtful suggestion, we have included additional loading controls, such as PARP and SKP1, and also Ponceau S (Figure 2C-D and EV1F). In Source data to corresponding figures we also included densitometry analysis of all studied proteins and Ponceau S stainings. While ARP3 is regulated during actin dynamics, it is occasionally employed as a loading control because its protein levels do not change during migration, cell cycle progression or cell growth. However, considering its involvement in cell dynamics, we recognize the importance of presenting a comprehensive set of loading controls. Importantly, actin, tubulin, and GAPDH, all highly expressed genes, have been shown to undergo regulation during immune response, T-cell activation, cell cycle, and migration. Therefore, we avoid frequent use of these markers. Additionally, in our analyses, we consistently verify precise loading through total protein staining with Ponceau S. We included Ponceau S stainings and densitometry analyses in our Source data.

FBXO38 ko mice: Could the authors please add IHC/IF of Fbxo38, PD-1 and ZXDB in immune cells of wild type and ko mice? This would further strengthen the reported observation.

We firmly believe that flow cytometry analysis of splenocytes is consistently the best quantitative assays for the analysis of protein-level of PD-1 in this tissue. We have included the flow cytometry analysis of permeabilized T cells to include the intracellular pool of PD-1. We observed very similar results to the surface PD-1 analysis consistently supporting the conclusion that FBXO38 does not regulate the levels of PD-1 in T cells (Figure EV1G-I).

FBXO38-PD-1 and cancer:

Meng et al. reported an involvement of FBXO38 in the T-cell mediated antitumour immunity. Could this be a discriminator for the biology of Fbxo38 towards PD-1 stability? Please clarify or discuss this point.

In our manuscript, we addressed the conclusion by Meng et al. that FBXO38 regulates PD-1 on the protein level. Their claim was based on observations in cell lines (their Figure 2) and primary T cells at the steady state and after ex vivo activation (their Figure 3A-C). They switched to the cancer model only in the subsequent experiments (Fig. 3D and beyond). Since our observations disproved that FBXO38 regulates PD-1 levels in various conditions (cancer cell lines, steady-state T cells, ex vivo activated T cells, and LCMV-activated T cells) where both PD-1 and FBXO38 are expressed, it is unlikely that such a regulation would appear only in the tumor setup. We also have doubts about some of the interpretations of Meng et al. about the role of FBXO38 in the anti-tumor immune response. Their conclusion “*The anti-tumour effect of IL-2 was evident in wild-type mice (Fig. 4m, n), stopping tumour growth in 2 out of 10 mice. This effect was apparently compromised in the Fbxo38CKO mice, showing that the anti-tumour function of IL-2 is partially dependent on FBXO38.*” can be made only if the data are presented as individual tumor growth (their Fig. 4N) without any quantification and statistical testing. In fact, the anti-tumor effect of IL-2 is apparent and quite comparable in WT and *Fbxo38* cKO mice, if we do quantify the tumor volumes (see Figure 1 below). The major difference between WT and *Fbxo38* cKO mice in this experiment seems to be the faster tumor growth in *Fbxo38* cKO mice, not the selective effect of IL-2 or dependency of the IL-2 therapy on FBXO38. However, this goes beyond the scope of our manuscript, as we focused solely on the alleged regulation of PD-1 by FBXO38 as mentioned above. We do not challenge the general observations by Meng et al. that FBXO38-deficient T cells are less efficient in the anti-tumor immunity, which was shown in multiple experiments (their Fig. 3D, 3M, 4N). However, we believe that, based on our study, the mechanism must not be via regulating the PD-1 levels by FBXO38. We clarified this in the revised version of the manuscript.

Figure 1. Reanalysis of data published by Meng et al. in their Figure 4N. It shows the size of B16F10 melanoma in WT and *Fbxo38* cKO mice treated or not with IL-2.

Overall, the presented manuscript is of high interest and touches on the regulation of a protein of high significance.

Once again, we would like to extend our sincere appreciation for this Reviewer's invaluable contribution to our manuscript.

References

1. Sakamoto, K.M. *et al.* Protacs: chimeric molecules that target proteins to the Skp1-Cullin-F box complex for ubiquitination and degradation. *Proc Natl Acad Sci U S A* **98**, 8554-8559 (2001).
2. Zhang, Z. *et al.* Elucidation of E3 ubiquitin ligase specificity through proteome-wide internal degron mapping. *Mol Cell* **83**, 4191-4192 (2023).
3. Latchman, Y.E. *et al.* PD-L1-deficient mice show that PD-L1 on T cells, antigen-presenting cells, and host tissues negatively regulates T cells. *Proc Natl Acad Sci U S A* **101**, 10691-10696 (2004).
4. Tit-Oon, P. *et al.* Intact mass analysis reveals the novel O-linked glycosylation on the stalk region of PD-1 protein. *Sci Rep* **13**, 9631 (2023).
5. Akcimen, F. *et al.* A novel homozygous FBXO38 variant causes an early-onset distal hereditary motor neuropathy type IID. *J Hum Genet* **64**, 1141-1144 (2019).
6. Sumner, C.J. *et al.* A dominant mutation in FBXO38 causes distal spinal muscular atrophy with calf predominance. *Am J Hum Genet* **93**, 976-983 (2013).
7. Grunseich, C. *et al.* Improving the efficacy of exome sequencing at a quaternary care referral centre: novel mutations, clinical presentations and diagnostic challenges in rare neurogenetic diseases. *J Neurol Neurosurg Psychiatry* **92**, 1186-1196 (2021).
8. Dibus, N. *et al.* FBXO38 Ubiquitin Ligase Controls Sertoli Cell Maturation. *Front Cell Dev Biol* **10**, 914053 (2022).

Dear Dr. Cermak

Thank you for the submission of your revised manuscript to EMBO reports. We have now received the report from the referee who was asked to assess it (copied below).

As you will see, the referee considers your response to the referee concerns adequate and supports publication.

Browsing through the manuscript myself, I noticed a few editorial things that we need before we can proceed with the official acceptance of your study.

- Your study will be published in our Reports section and therefore the Results and Discussion sections should be combined. That said, I do not recommend to shorten the Discussion.
- Please include the link to the additional source data deposited on BioStudies in the Data Availability Section.
- Please change the header 'Conflict of interest' to 'Disclosure and competing interests statement'.
- Please remove the Author Contributions from the manuscript file and make sure that the author contributions in our online submission system are correct and up-to-date. The information you specified in the system will be automatically retrieved and typeset into the article. You can enter additional information in the free text box provided, if you wish.
- Please link the ORCID ID of Dr. Pagano to the account in our manuscript tracking system. All corresponding authors are required to supply an ORCID ID.
- Information on funding needs to be part of the Acknowledgments section. Moreover, all funding must be specified in the manuscript tracking system. In this regard we note that grant # 260637 is missing in the system.
- Appendix Table S1: if this table/figure should be part of the Appendix, please label it as Appendix Table S1 and provide it in PDF format with a title page that includes a table of content with page numbers (even if it is only one page). If you prefer to keep the table in .xls format, it could be Table EV1. Alternatively, you provide it as figure/table in the Appendix and the raw data as Source Data for the Appendix. That might be the most transparent and useful option.
- Author Checklist: please complete the part on Dual Use Research of Concern (D99, D100 and D101) by choosing a response from the drop-down menu.
- We can only typeset one Reagents and Tools table in the manuscript. I suggest to supply the .xls Reagent table as Table EV1. Please update the entry in the Author Checklist accordingly.
- The correct reference to the Reagents table is "Reagents and Tools table". Please correct the callout in the methods section (cDNA preparation and RT-qPCR analysis).
- The methods section should be renamed. It is called "Materials and Methods" with subheadings "Reagents and Tools table" followed by "Methods and Protocols".
- The Ethics Statement should be part of Materials and Methods.
- Figure EV1 consists of two uploaded files EV1a and EV1b. Please either merge these into one file or rename them to Figure EV1 and Figure EV2 (preferred option).
- The manuscript sections should be in the following order: Title page - Abstract & Keywords - Introduction - Results - Discussion - Materials & Methods - Data Availability - Acknowledgments - Disclosure Statement & Competing Interests - References - Figure Legends - Tables with legends - Expanded View Figure Legends.
- On page 5 you refer to 'Expanded Fig 3 A,B' which seems wrong. Please check.
- Please note that you can cite datasets in addition to the paper reporting on it if you reuse publicly available data (e.g., Ogando et al, 2019, Yukawa et al 2020). If you use data citations, you cite both, the paper and the dataset. In the example below the citation would look like (Smith et al, 2001, Data ref: Smith et al, 2001).
Data citations in the article text are distinct from normal bibliographical citations and should directly link to the database records from which the data can be accessed. In the main text, data citations are formatted as follows: "Data ref: Smith et al, 2001" or "Data ref: NCBI Sequence Read Archive PRJNA342805, 2017". In the Reference list, data citations must be labeled with "[DATASET]". A data reference must provide the database name, accession number/identifiers and a resolvable link to the

landing page from which the data can be accessed at the end of the reference. Further instructions are available at <<https://www.embopress.org/page/journal/14693178/authorguide#referencesformat>>.

- Please check the source data for Figure 1E again. In the folder Fig_1E_roi the source data do not always match. E.g., the file labeled '1E_MG132_DOX-subset' matches the panel '+DOX +DMSO'. The file '1E_DMSO_DOX_subset' matches the panel '-DOX +DMSO', the file 'DMSO_NO_DOX' seems not to match any panel in Figure 1E. 'MG132_DOX_subset' matches '+DOX +DMSO' etc. The raw data 'lif' files match the panels.

- Figure 2D: the source data for the PARP blot is missing.

- Source data for Figure 3G: you present the Ct values normalized to Tbp, but should the Tbp Ct values not differ between the different samples?

- For completeness, I recommend uploading the fcs source data files for Figure EV1 and EV2 to Biostudies as well.

- Finally, EMBO Reports papers are accompanied online by A) a short (1-2 sentences) summary of the findings and their significance, B) 2-3 bullet points highlighting key results and C) a synopsis image that is 550x300-600 pixels large (width x height) in PNG or JPG format. You can either show a model or key data in the synopsis image. Please note that the size is rather small and that text needs to be readable at the final size. Please send us this information along with the revised manuscript.

Kind regards,

Referee #2:

Reviewer #2 final report:

The authors have done an excellent job of addressing my questions and I think the manuscript is now suitable for publication. The additional experiments performed in response to all the reviewers question have significantly improved on their original story.

The authors addressed the minor editorial issues.

Lukas Cermak
Institute of Molecular Genetics CAS
Cancer Biology Lab
Videnska 1083
Praha 14220
Czech Republic

Dear Lukas,

Thank you for your patience while we have been waiting for the referee's feedback on the related Correspondence article. I am now pleased to accept your manuscript for publication in the next available issue of EMBO reports. Thank you for your contribution to our journal.

Kind regards,

Martina
